# Regional coupled surface-subsurface hydrological model fitting based on a spatially distributed minimalist reduction of frequency-domain discharge data

Nicolas Flipo[1], Nicolas Gallois[1], and Jonathan Schuite[2,1]

[1]Mines Paris, PSL University, Centre for geosciences and geoengineering, 77300 Fontainebleau, France
[2]TerraScience, 24 120 Terrasson-Lavilledieu, France

**Correspondence:** Nicolas Flipo (nicolas.flipo@minesparis.psl.eu)

**Abstract.** Although integrated water resources models are indispensable tools for water management at various scales, it is of primary importance to ensure their proper fitting on hydrological variables, avoiding flaws related to equifinality. An innovative step-wise fitting methodology is therefore proposed, which can be applied for any river basin model, from catchment to continental scale as far as hydrological models or land surface models are concerned. The methodology focuses on hydrosystems considering both surface water and groundwater, as well as internal water fluxes such as river baseflow. It is based on the thorough analysis of hydrological signal transformation by various components of a coupled surface–subsurface hydrosystem, in a nested approach, that considers the conditionality of parameter fields on their input forcing fluxes.

The methodology is based on the decomposition of hydrological signal in the frequency domain with the HYMIT (HYdrological MInimalist Transfer function) method (Schuite et al., 2019). Parameters derived from HYMIT are used to fit the coupled surface–subsurface hydrological model CaWaQS3.02 using a step-wise methodology, which relies on successive Markov chain Monte Carlo optimizations, related to various objective functions representing the dependency of the hydrological parameters fields on forcing input fluxes. This new methodology enables significant progress to be made in terms of the spatial distribution of the model parameters and the water balance components, at the regional scale. The use of many control stations such as discharge gauging stations with HYMIT leads to a coarse parameter distribution that is then refined by the fitting of CaWaQS parameters on its own mesh.

The step-wise methodology is exemplified with the Seine River basin ($\sim$ 76,000 $km^2$). In particular, it made it possible to spatially identify fundamental hydrological values, such as rainfall partitioning into actual evapotranspiration, as well as runoff and aquifer recharge through its impluvium, in both the time and frequency domains. Such a fitted model facilitates the analysis of both the overall and detailed territorial functioning of the river basin, including explicitly the aquifer system. A reference piezometric map of the upmost free aquifer units and a water budget of the Seine basin are established, detailing all external and internal fluxes up to the exchanges between the eight simulated aquifer layers. The results showed that the overall contribution of the aquifer system to the river discharge of the river network in the Seine basin varies spatially within a wide range (5–96 %), with an overall contribution at the outlet of the basin of 67 %. The geological substratum greatly influences the contribution of groundwater to the river discharge.

# 1 Introduction

Given the current climate and anthropogenic trends, water management has become one of the greatest challenges of the $21^{st}$ century. Today, less than 1% of the global water stock is readily available for human activities, among which only a fraction constitutes a renewable water stock (Oki and Kanae, 2006; Roche and Zimmer, 2006; de Marsily, 2008). Overall, the pressure on the earth hydrosystem is such that approximately 500 M people are already experiencing water stress throughout the year (Mekonnen and Hoekstra, 2016). Although uncertainties remain regarding the quantification of climate change impacts on water resources (Taylor et al., 2013), the number of people exposed to hydrological stresses will rise in the near future. The main source of uncertainties for such predictions is related to climate models, but uncertainties related to hydrological models are not negligible (Hattermann et al., 2018; Her et al., 2019; Ashraf Vaghefi et al., 2019). It is therefore important to ensure the proper adjustment of these models to hydrological data.

In the Anthropocene Epoch (Crutzen, 2002; Crutzen and Steffen, 2003), it is not only climate that affects the trajectory of large river basins but also human pressure, such as land cover changes and water withdrawal that can sometimes become the controlling factors of those systems, overcoming natural factors (Rockström et al., 2009). In some regions of the world, water demand for sociological purposes will also drive water cycle changes in the same order of magnitude as climate change could affect the system (Haddeland et al., 2014). It is therefore now acknowledged that integrated modeling tools at the regional basin scale are the most suited for water management and planning purposes (Perkins and Sophocleous, 1999; Flipo et al., 2014; Barthel and Banzhaf, 2016), usually based on scenario testing (Hattermann et al., 2017), as well as for disentangling climate change impacts on the water cycle from a growing water demand associated with population growth (Jackson et al., 2001; Flipo et al., 2021a). In this context, improvements in hydrological modeling tools are needed in order to properly estimate the spatio-temporal dynamics of water fluxes involved in the terrestrial water cycle (Uniyal et al., 2015). Regarding climate impact studies, it is of utmost importance to ensure the proper calibration of hydrological models before using them as prospective tools.

Since the hydrosystem modeling blueprint proposed by Freeze and Harlan (1969); Simmons et al. (2020), and efficiently implemented in many well-known models, hydrology and hydrogeology communities alike have followed divergent paths leading to a paradoxical situation where even though each discipline depends on the other through boundary conditions, collaborative work between both sides needs to be reinforced (Staudinger et al., 2019). As a concrete consequence of this situation, modelers are torn apart between more hydrological interests, usually at the continental scale where the calibration of the models is mostly focused on discharge, and catchment or regional scale where occasionally hydrogeologists add hydraulic head to the fitting process through reformulation of objective functions (Saleh et al., 2011; Flipo et al., 2012; Pryet et al., 2015; Baratelli et al., 2016). All these approaches suffer from large uncertainties in the identification of parameters known as "equifinality" (Beven and Binley, 1992; Beven, 2006; Ebel and Loague, 2006). On the one hand, fitting model parameters, and especially groundwater (GW) models, on discharge data does not prove neither that the model reproduces the correct physical processes, nor that the distribution of river–aquifer exchanges is correctly located along a river network at the watershed scale (Barclay et al., 2020). On the other hand, the most recent benchmarking strategies rely on the ability of the model to reproduce physical

processes based on simplistic case studies but not on a data–model comparison strategies (Maxwell et al., 2014; Tijerina et al., 2021) that overcome equifinality issues.

From the models' review of Singh and Woolhiser (2002),Paniconi and Putti (2015) and Fatichi et al. (2016), three main types of fully integrated model structures can be distinguished: **(i)** fully physically-based 3-D models – *e.g.* Cast3M (Weill et al., 2009), CatHy (Camporese et al., 2010)), Hydrogeosphere (Therrien et al., 2010), InHM (VanderKwaak and Loague, 2001; Loague et al., 2005), MODHMS (Panday and Huyakorn, 2004), OpenGeoSys (Kolditz et al., 2012), ParFlow (Kollet and Zlotnik, 2003), PIHM (Qu and Duffy, 2007) – **(ii)** fully physically based pseudo 3-D models – *e.g.*, PAWS (Shen and Phanikumar, 2010), Mike-she (Abbott et al., 1986a, b) – or **(iii)** coupled conceptual–physically-based models – *e.g.*, CAWAQS-like (Flipo et al., 2021a), GSFLOW (Markstrom et al., 2008) –, for which surface processes are simulated using conceptual reservoir models. They all compute the hydrological processes controlling hydrosystem[1] functioning. They are thus particularly suited for reporting the spatial and temporal dynamics of water fluxes for water management purposes (Labarthe et al., 2015). However, results provided by such integrated models can be uncertain (Wu et al., 2014). These issues can be due to the significant number of calibration parameters involved (Wu et al., 2014) and to the reliance of subsurface parameters on recharge rates estimated by simulation of surface processes (Erdal and Cirpka, 2016). In order to reduce integrated model uncertainties, their parameters need to be defined more precisely, through specific calibration procedures. In this sense, Flipo et al. (2012) introduced a step-wise calibration strategy of hydrosystem models in which surface and subsurface parameters of hydrosystem models are calibrated in a sequential fashion to address their dependency. In this procedure, surface and subsurface models are iteratively optimized until the calibrated parameter set reproduces both observed groundwater levels and river discharges. This procedure introduces the fluxes occurring at the surface–subsurface interface (aquifer recharge and stream–aquifer interactions) in the calibration procedure indirectly and accounts for the dependence of subsurface parameters on surface recharge. However, even if the step-wise calibration strategy has proven its efficiency in fully coupled model calibration (Flipo et al., 2012; Pryet et al., 2015; Baratelli et al., 2016), some aspects remain critical such as the computational burden of conducting the iterative procedure and the potential bias in the simulation of the water budget.

In this paper, we demonstrate a step-wise methodology that builds on Flipo et al. (2012). It considers the conditionality of GW parameters on their boundary conditions and relies on estimates of internal hydrosystem flux values to improve the calibration of spatialized hydrosystem parameters. Besides bringing together the interests of the two communities of hydrologists and hydrogeologists, the methodology intends to significantly reduce the equifinality issue related to the fitting of simulation models of hydrosystems, in terms of the watershed. The methodology can be applied for any river basin model from the catchment scale to the continental scale as far as Land Surface Models (LSM, Pitman (2003)) are concerned, such as CLM (Lawrence et al., 2019; O'Neill et al., 2021), MGB–IPH (Collischonn et al., 2007; Paiva et al., 2013), ORCHIDEE (Ducharne et al., 2003; Krinner et al., 2005), SURFEX (Masson et al., 2013; Le Moigne et al., 2020), VIC (Liang et al., 1994, 1996; Hamman et al., 2018), or hydrological models of increasing complexity such as for instance GR (Perrin et al., 2003) or mHM (Samaniego et al., 2010). The methodology focuses on hydrosystems considering both surface water and GW, as well as internal water

---

[1] "Hydrosystem" is used here following the definition of Flipo et al. (2012), which corresponds to a whole river basin, where both surface and groundwater are accounted for.

fluxes such as river baseflow. It is based on the thorough analysis of hydrological signal transformation by various components of a hydrosystem in a nested approach, which takes into account the conditionality of parameter fields on their input forcing fluxes. It develops further the fact that regional basins behave as low-pass filters due to the effect of large scale aquifer systems (Baulon et al., 2022b, a) and the potential of aquifer parameter identification with spectral analysis of *in situ* piezometric data versus an estimate of the aquifer recharge (Houben et al., 2022).

The methodology, described in detail in section 2, is based on the decomposition of a hydrological signal in the Fourier frequency domain with the HYMIT (HYdrological MInimalist Transfer function) method (Schuite et al., 2019; Schuite, 2022). HYMIT makes it possible to study the influence of the physical properties of watersheds on the deformation of precipitation signals. It is based on a transformation of hydro-meteorological data in the Fourier frequency domain, in which a transfer function is composed of a minimalist number of hydrological processes and parameters are fitted with a Markov chain Monte Carlo (MCMC) approach. Parameters derived from HYMIT are used to fit the coupled surface–subsurface hydrological model CaWaQS3.02 with a step-wise approach, which relies on successive MCMC optimizations, respectively, related to various objective functions that represent the dependency of hydrological parameter fields on forcing input fluxes.

While section 3 explores the performance of the model from both quantitative and qualitative perspectives, section 4 illustrates the significant progress in terms of spatialization of the water balance components, as well as stream–aquifer exchanges, at the regional scale, enabled by the proposed methodology. It is exemplified with the Seine River basin ($\sim$ 76,000 $km^2$). More specifically, these two sections reveal the consistency of the spatially distributed physical properties of the Seine sub-catchments, derived from the reproduction of the partitioning of effective rainfall into fast surface flow and slow flow in the subsurface domain, with morphological data and geomatic analyses. They highlight the evolution of the filtering effect of precipitation signals by successive catchments from upstream to downstream at the scale of the Seine basin. Finally, a discussion (section 5) points out the relevance of the developed methodology for hydro(geo)logical modeling in general, from the catchment to the continental scale.

## 2  Material and method: Fitting a regional river basin hydrosystem model

Fitting a regional hydrosystem model, which couples fast surface and slow subsurface flows at the scale of a regional basin, such as the Seine River basin, is a challenging issue (Flipo et al., 2012), especially regarding GW, since measurements of the fluxes are unavailable.

Although automatic adjustment methods do exist in disciplinary fields, whether in hydrology or hydrogeology, this is not the case when one is interested in the coupled dynamics of surface and underground processes. In their review of this topic, Flipo et al. (2012) proposed to circumvent the problem using a nested loop fitting method. The basis of the method is to accept the conditionality of GW flows on the process of aquifer recharge through its impluvium. It is then possible to nest two loops: one dealing with fast surface processes and the other associated with slow subsurface processes. An essential condition for the implementation of this method is that the first loop enables the quantification of the forcing of the second loop, *i.e.*, recharge of aquifers by their impluvium.

An initial implementation of this two-step methodology, each one being fully automated, was proposed by Labarthe (2016), and has been used to reconstruct the hydrological trajectory of the Seine basin since the beginning of the XX$^{th}$ century (Flipo et al., 2021a). This method, relying on the fact that, for a 17-year stationarity period, river baseflows can be equated to the aquifer recharge across the river basin, depends on additional information to the classic datasets of river discharges and groundwater hydraulic heads, namely, an estimate of the river baseflow. However, these estimates are still marred by an
error that we acknowledge, but that we are unable to quantify, mainly due to the great difficulty (or even impossibility) of the measurement at large scale.

We therefore develop a new fitting methodology based on a thorough analysis of hydrological signal transformation by various components of a hydrosystem in a nested approach, which considers the conditionality of parameter fields on their input forcing fluxes. The approach is based on the decomposition of hydrological signals in the Fourier frequency domain with
the HYMIT method (Schuite et al., 2019).

## 2.1 The Seine River basin

### 2.1.1 A highly anthropized river basin

The Seine River basin ($\simeq 76{,}000\ km^2$) is a highly human-impacted basin with a population of 17 M inhabitants. It receives the highest anthropogenic pressure in France, due to the industry and agriculture linked to the development of the urban area of
Paris, which coexists today with highly productive agricultural areas (66 % of the basin area, Fig 1). For more information on the Seine basin, please refer to Billen et al. (2007) and Flipo et al. (2021b).

The river network is composed of 28,000 km of perennial rivers (Fig. 2). Overall, 97 % of the river network lies within the sedimentary Paris basin (Guillocheau et al., 2000), the largest GW reservoir in Europe. The inter-annual mean values (2003–2020 period) of the climate forcing (SAFRAN database – Quintana-Seguí et al. (2008)) are $766\ mm\,a^{-1}$ and $890\ mm\,a^{-1}$ for
precipitation and potential evapotranspiration (PET), respectively. The inter-annual module of the Seine river at Vernon (Fig. 2) is $477\ m^3\,s^{-1}$ (2010–2019, HYDRO database daily dataset).

Water withdrawals in surface water and GW amount to $3\ km^3\,a^{-1}$. The pressure on water resources is high especially during low-flow periods, when river discharges are sustained by GW and by human-built reservoirs (Fig. 2), totalling $841\ 10^6\ m^3$.

### 2.1.2 Naturalization of downstream discharges from reservoirs

Here, observed river discharges are studied as an overall integrator of the system's behavior and response to climatic fluctuations. In this context, preliminary corrections are required to ensure the absence of anthropogenic disturbances in measured data that might result from the significant pressure the basin is submitted to. Since the HYMIT approach is not designed to dissociate such perturbations in measured discharge variations, the imprints left in the observed data by water reservoir storage/release cycles have been subtracted from downstream station records. Two types of information were used to perform this
discharge naturalization: **(i)** reservoir withdrawn or released volumes to the river system and **(ii)** water travel times along the network, respectively, associated with each downstream station from one or several reservoirs.

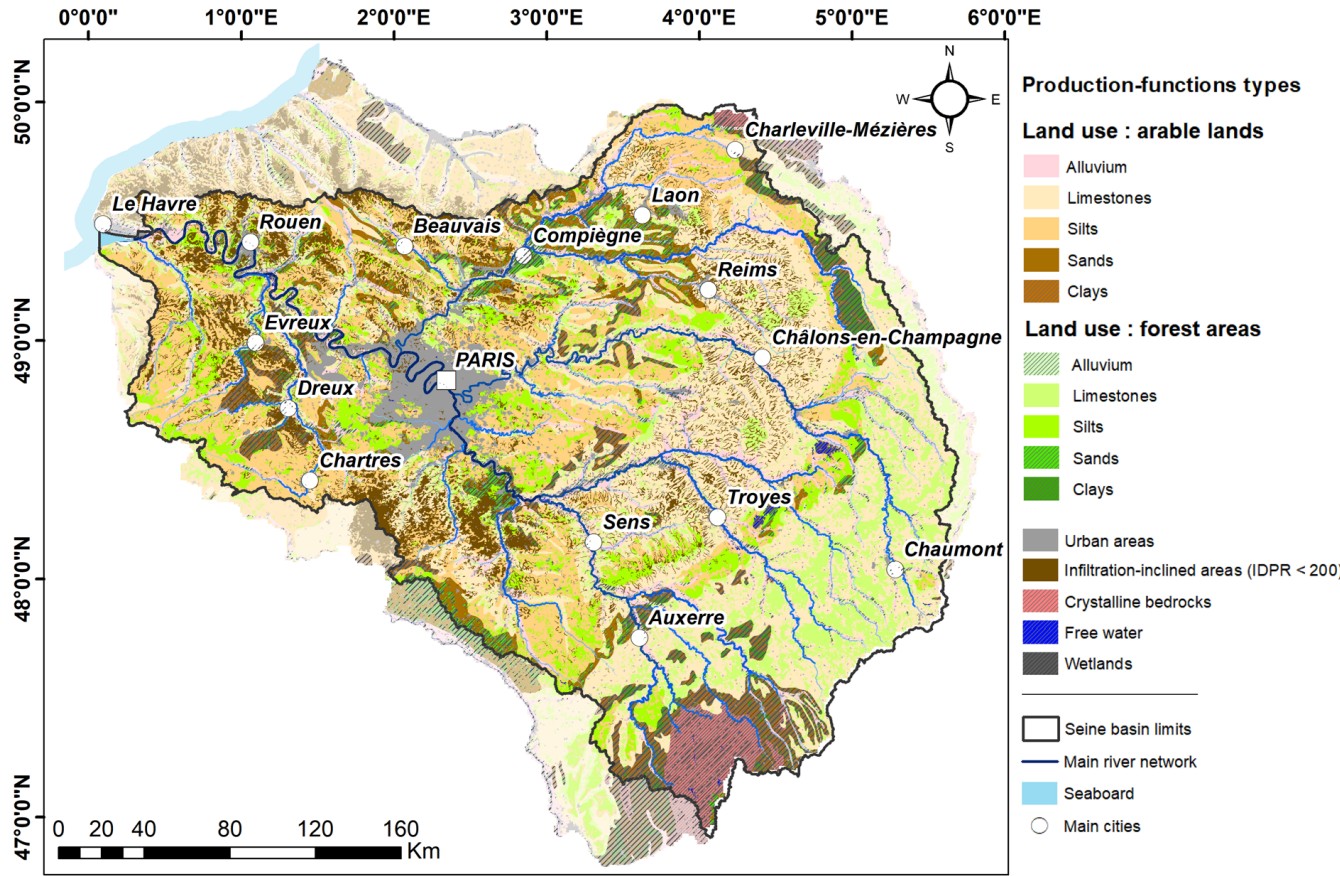

**Figure 1.** Joint representation of land use and dominant lithology through production function distribution over the modeled area. Hatched zones highlight functions with lower minority share in total surface (*i.e.* $\leq 2\,\%$), which are excluded from the HYMIT–MCMC analysis.

The calculation of a naturalized discharge value $Q^i_{nat}(t)\,[m^3\,s^{-1}]$ associated with its respective actual measurement $Q^i_{mes}(t)$ $[m^3\,s^{-1}]$ is performed using equation (1), written in the case of a station $i$, at a time $t$, located downstream from $N$ reservoirs:

$$Q^i_{nat}(t) = Q^i_{mes}(t) + \sum_{k=1}^{N} Q^i_{dam,k}\left(t - T^i_{tra,k}\right) \tag{1}$$

where $Q^i_{dam,k}(t - T^i_{tra,k})\,[m^3\,s^{-1}]$ represents the daily volume either stored from ($>0$) or released to ($<0$) the river system, accounting for the water travel time $T^i_{tra,k}\,[s]$ along the network fraction between dam $k$ and station $i$.

Travel times are calculated for each reach $r$ of the network, using a relative transfer time index $I_{tr}(r) = dl(r)/\left(\sqrt{s(r)}\,S(r)^\gamma\right)$, as a function of geomorphological data (Golaz-Cavazzi, 1999; Flipo et al., 2012):

– the distance $dl(r)\,[m]$ between center of reach $r$ and its contiguous downstream reach;

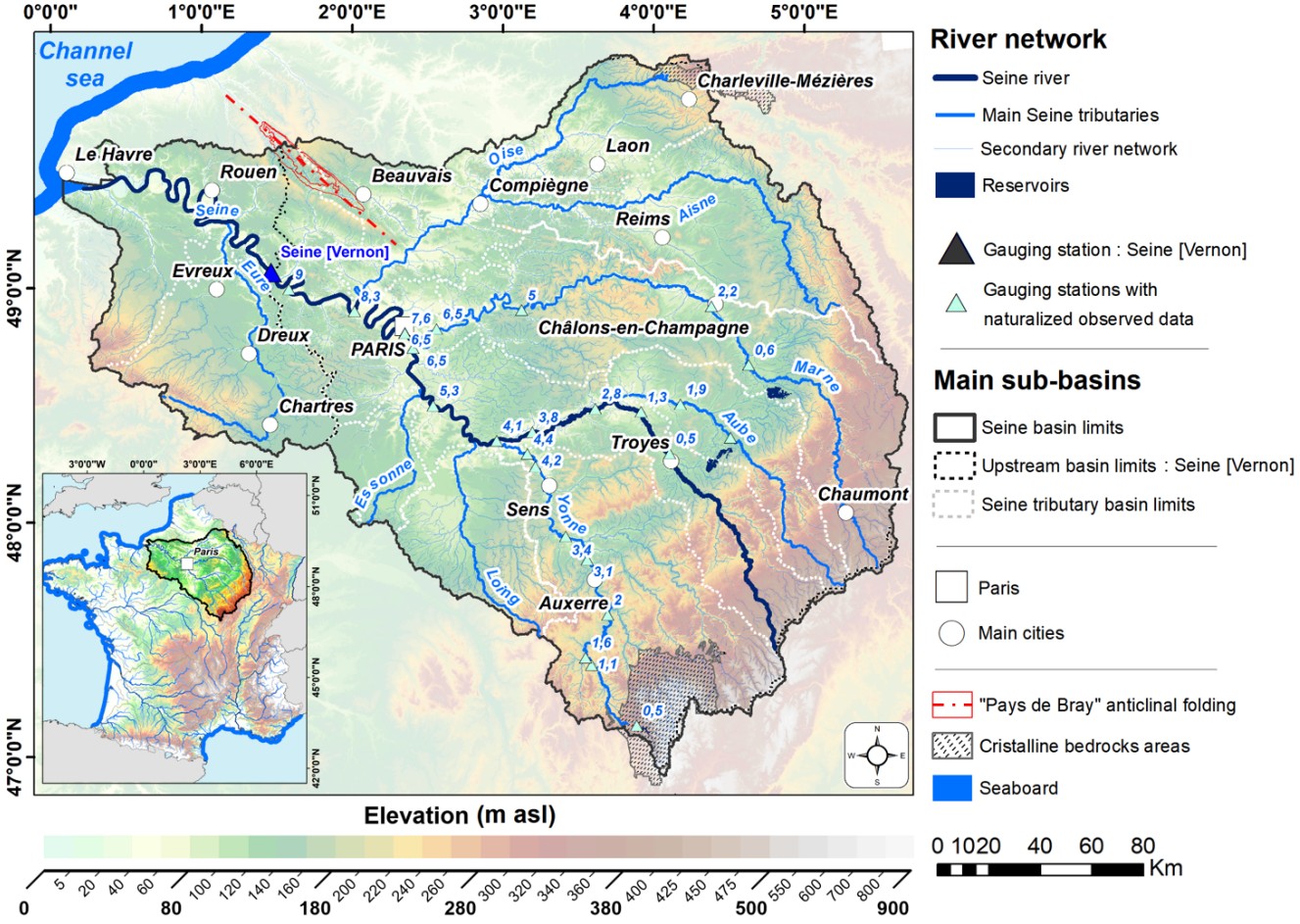

**Figure 2.** General overview of the Seine basin area. 25m-resolution DEM is used as map background. Light-blue labels refer to travel times (in days), from reservoirs to downstream discharge gauging stations. $asl$ : Above Sea Level.

– reach slope $s(r)$ $[-]$, derived from DEM;

– the cumulative reach upstream drainage area $S(r)$ $[km^2]$ and a calibration parameter $\gamma = 0.25$ (Korkmaz, 2007).

For every reach, a relative transfer time index to the basin outlet $I_{tr;r \to out}$ is calculated as a sum of local $I_{tr}$ along all reaches leading to the outlet. A travel time $T_{tra,k}^i$, is then calculated following eq. 2:

$$T_{tra,k}^i = \left( \frac{I_{tr,k \to out} - I_{tr,i \to out}}{I_{tr,max}} \right) T_c \tag{2}$$

Where $I_{tr;max}$ is the maximum relative time index to the outlet basin and $T_c$ $[s]$ is the basin global concentration time, considered equal to 17 days (Gomez, 2002; Saleh et al., 2011; Pryet et al., 2015).

Naturalization of discharge records was carried out for the 30 measurement sites mentioned in figure 2, for which travel time values from reservoirs to stations (in days) are also displayed. In the case of several upstream reservoirs, labels correspond to mean travel time values.

## 2.2 The physically based coupled surface–subsurface model CaWaQS3.02

The physically-based CaWaQS coupled model (CAtchment WAter Quality Simulator – Flipo et al. (2005, 2007b, a, 2021a); Labarthe (2016)) was used to: **(i)** estimate the distributed effective rainfall over the basin as an input of a distributed HYMIT analysis at 221 river discharge gauging stations, **(ii)** estimate the basin physical parameters, and **(iii)** model the Seine basin pluri-decennial functioning. Based on the blueprint first published by de Marsily et al. (1978) and implemented as the MODCOU–NEWSAM software suite (Ledoux et al., 1984, 1989), CaWaQS3.02 (Flipo et al., 2022b) is a spatially distributed model that simulates coupled water, matter, and energy balances and flow dynamics within all compartments of a hydrosystem. The software structure links dedicated C-ANSI libraries meant to mimic main physical processes controlling the fate of water in each compartment. Calculations of surface, subsurface, and GW dynamics involve five main modules using a daily time-step (Labarthe, 2016) :

– a surface module (Fig. 3$a$), which computes estimates of AET, runoff, and infiltration fluxes (see Fig. A1 in appendix for more details). Relying on a conceptual reservoir-based approach (Girard et al., 1980; Deschesnes et al., 1985), water balance calculations account for climate data (see 'total rainfall' and PET maps in Figs. 9$a$ and 9$b$, respectively) as well as the distributions of land use and parent soil material (Fig. 1). Runoff water production is aggregated according to local subcatchments to ensure its direct transfer to the river system. These catchments can also be set up as runoff short-circuits toward the unsaturated zone ("chasm"-type configuration). Details on the surface module are available as supplementary material with a special emphasis on AET calculation;

– a vadose zone module (Fig. 3$c$), which vertically transfers infiltration from the surface domain to subsurface outcropping aquifer areas. It diffuses soil infiltration based on a Nash reservoir cascade (Nash, 1959; Besbes and De Marsily, 1984) toward the aquifer system, which is equivalent to a gamma distribution function. This approach is an efficient and performing alternative to the Richards formulation of water flow in an unsaturated porous media below the root zone (Besbes and De Marsily, 1984). It is particularly well adapted at the regional scale for analysing groundwater level fluctuations (see for instance the recent studies of Cao et al. (2016); Jeong et al. (2018); Park et al. (2021)), when preferential flow paths (Mirus and Nimmo, 2013; Nimmo, 2020a, b; Nimmo et al., 2021) are negligible, even though 10 to 40 % of very deep aquifer recharge can originate from young water flowing through those preferential flow paths (Jackson et al., 2022);

– a groundwater or aquifer system module (Fig. 3$d$), based on the pseudo-3-D diffusivity equation (de Marsily, 1986), solved using a semi-implicit finite volume numerical scheme, applied on nested grids. Besides integrating water recharge and anthropogenic withdrawals, it accounts for both confined- and unconfined-related resolution particularities, and also handles, along time and space, reversible transitions between these two states. Exchanges between aquifer units are

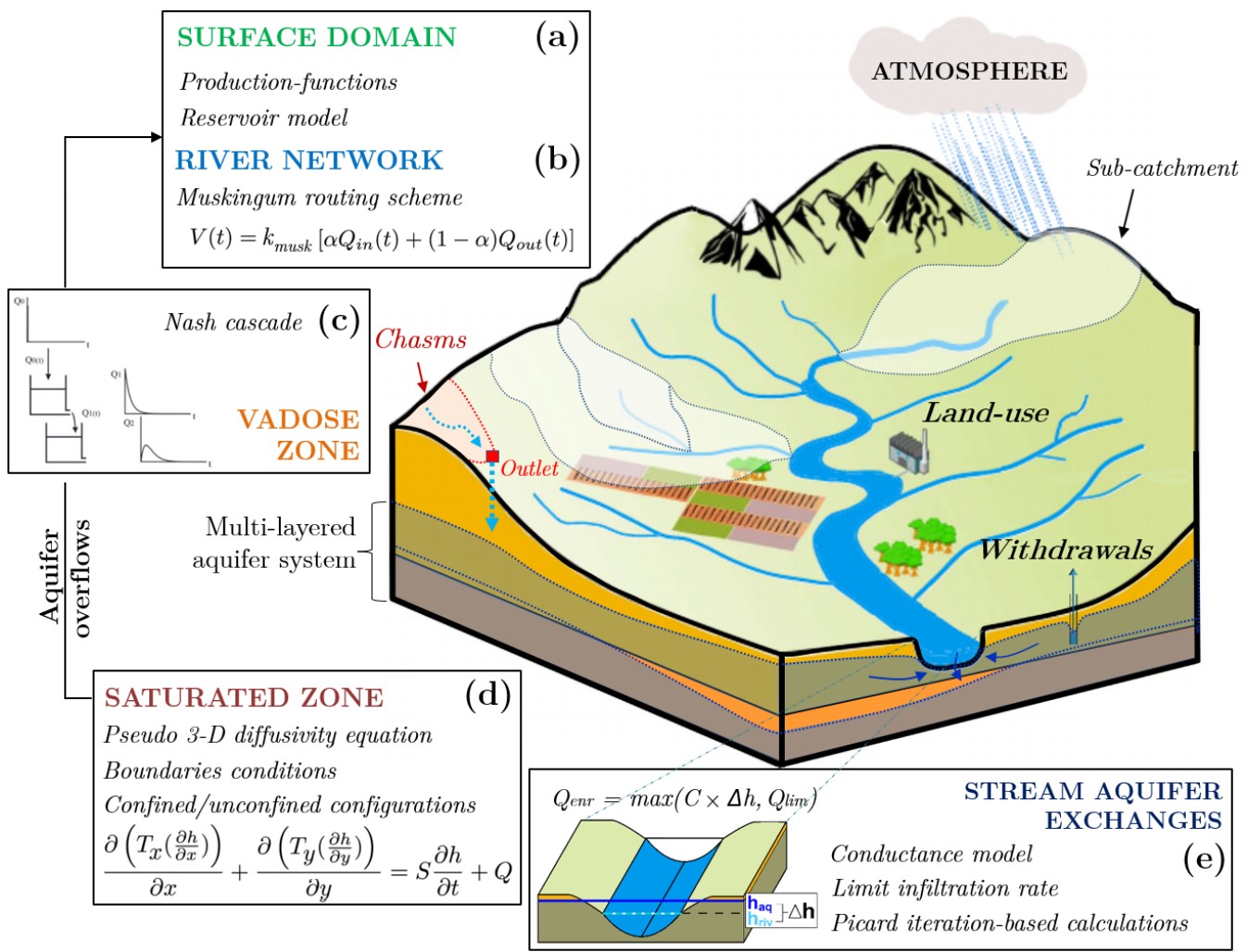

**Figure 3.** Simplified illustration of the structure of the CaWaQS3.02 hydrosystem modeling platform and its main concepts involved in water flow calculations. Notations : ($b$ : River network) $\alpha$: a weighting parameter ($\in [0;1]$), $k_{musk}$: transfer time between two adjacent river elements, ($Q_{in}$, $Q_{out}$): river element input and output discharges, $V$: water volume contained in a river calculation element. ($d$ : Saturated zone) ($T_x$, $T_y$): transmissivity coefficients in $x$ and $y$ directions, $S$: storage coefficient, $Q$: source term. ($e$ : Stream–aquifer exchanges) $C$: conductance coefficient, $Q_{enr}$: stream–aquifer exchanges flow, $Q_{lim}$: river-to-aquifer limit infiltration rate, $\Delta h$: difference between water height in river ($h_{riv}$) and aquifer hydraulic head ($h_{aq}$).

simulated based on a 1-D vertical simplification of water fluxes, which are assumed to be linearly connected to the head difference between aquifer units;

- a non-linear conductance model (Fig. 3e), which accounts for a limitation of the infiltration flux in the case of disconnection (Brunner et al., 2009; Rivière et al., 2014; Newcomer et al., 2016), is integrated within a Picard-iterative approach to compute stream–aquifer exchanges (Rushton, 2007; Ebel et al., 2009; Flipo et al., 2014);

- a hydraulic module (Fig. 3b), which transfers in-stream water discharges using a Muskingum routing scheme (David et al., 2011, 2013). For each river network cell, computed discharges integrate stream–aquifer fluxes, inputs due to subsurface runoff as well as exogenous point injection flows.

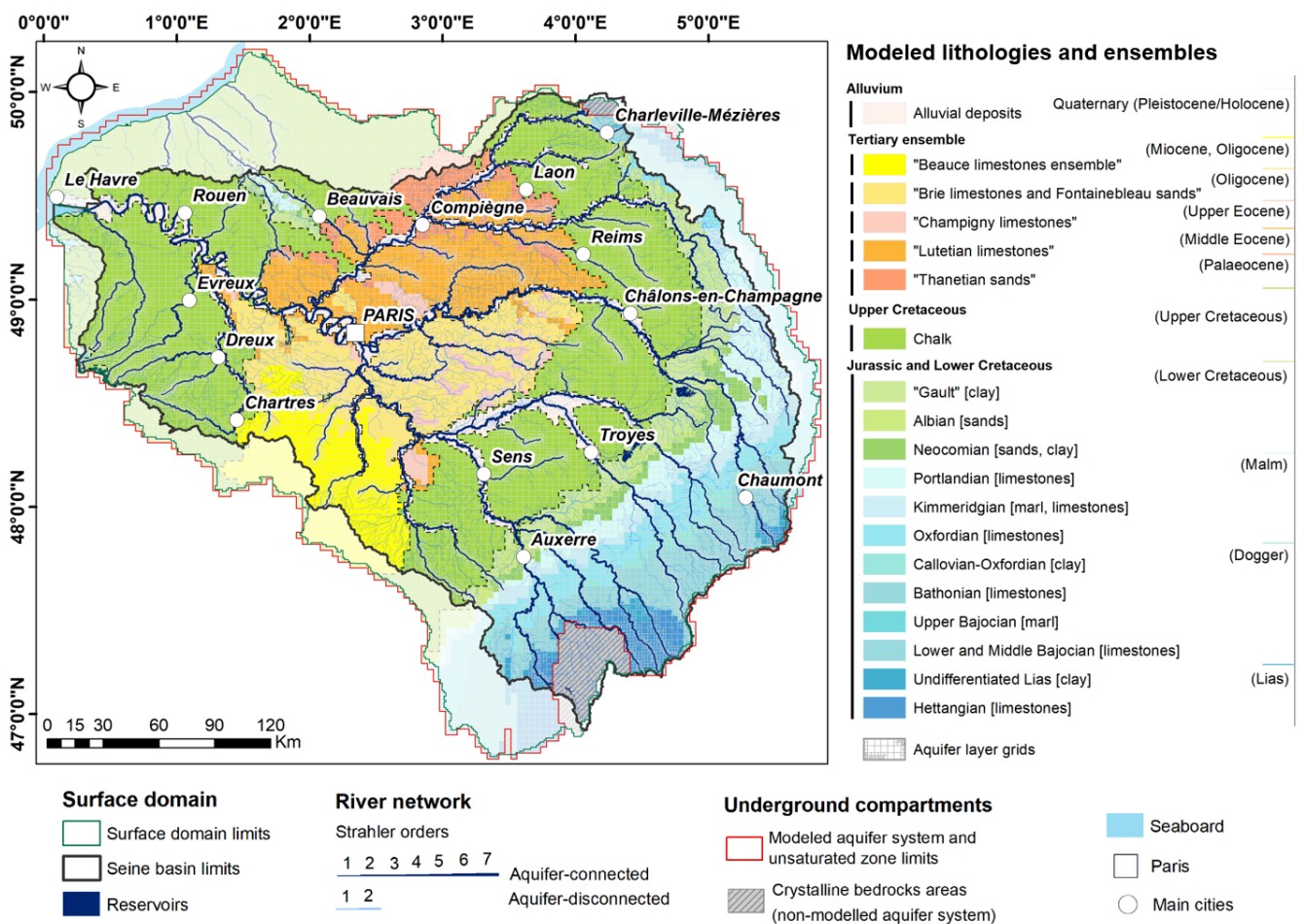

**Figure 4.** CaWaQS–Seine application structure overview. Modeled lithologies are gathered according to the main ensemble they belong to. Ensembles limits are delineated on the map using dashed lines. Where appropriate, the most common formation names are used using quotation marks, and dominant lithology types are mentioned between square brackets.

Initially designed by Gomez et al. (2003) and improved later on by Pryet et al. (2015); Labarthe (2016); Baratelli et al. (2018); Flipo et al. (2021a), the Seine basin application accounts for:

- a surface layer ($\simeq$ 95,100 $km^2$), divided into elementary calculation cells of 11 $km^2$ in average size, which covers the entire Seine basin;

- a river network that corresponds to 6,830 km of rivers. Mainly due to computational time concerns, calculations of stream–aquifer exchanges have been constrained to rivers including reaches with Strahler orders (Strahler, 1957) from 3 to 7 or 8 depending on the definition of perennial rivers in the database used to define the river network;

- a multi-layered aquifer system divided into 20 lithology units. These units are meshed using multi-scale nested grids with square-shaped cells ranging from 3,200 to 100 $m$ in size. From oldest to most recent, these units can be regrouped into four main categories (Fig. 4) : **(i)** an alternating ensemble of aquifer and aquitard units, mostly made of limestone and marl–clay associations, respectively. As a whole, they range from Lower Jurassic (Hettangian stage,- 195 $Myr$) to Lower Cretaceous (Albian stage, - 100 $Myr$) and mostly outcrop on the eastern end of the basin, **(ii)** a large Upper Cretaceous chalk layer, **(iii)** a 5-layer Tertiary complex ensemble, located in the center of the basin, which covers units mainly made of limestone and sand, dating back from the Paleocene to Miocene stages, and **(iv)** recent alluvial deposits (Pleistocene and Holocene stages, from -2.5 $Myr$ to -10,000 years, respectively) surrounding the main rivers. Areas where crystalline bedrock outcrops (Morvan, Ardennes) are not explicitly simulated ($\simeq$ 1.9 % of total modeled surface) (Fig. 2).

## 2.3 Minimalist reduction of frequency domain hydrological data with HYMIT

The HYMIT (HYdrological MInimalist Transfer function) method was designed by Schuite et al. (2019) and Schuite (2022) to describe how hydrosystems transform a climatic input signal, namely, effective rainfall, into observed hydrological responses such as river discharges or GW levels. Based on previous theoretical developments in the field of stochastic hydrology and frequency-domain analysis of hydrological variables (Gelhar, 1974; Molénat et al., 1999; Russian et al., 2013), HYMIT features a complete yet simple characterization of the filtering effects on flow dynamics by the three main compartments comprised in a hydrosystem: the surface (runoff, taking into account overland and hypodermic flow), the unsaturated porous subsurface (vadose zone), and the saturated subsurface (aquifer system). In other words, it links the expression of multi-frequency climate variability in GW levels and river discharges to the hydraulic and hydrogeological properties of hydrosystems through a transfer function analysis in the frequency-domain. In practice, HYMIT is adjusted to experimental transfer functions built from the discrete Fourier transforms of effective rainfall and river discharge time series. Therewith, it is possible to rapidly obtain a first-hand estimation of key watershed properties, by taking advantage of the full statistical power of long time series and the tractable analytical description of a catchment's hydrological functioning (Pedretti et al., 2016; Jiménez-Martínez et al., 2013; Jimenez-Martinez et al., 2016; Manga, 1999; Molénat et al., 1999; Schuite et al., 2019).

The ratio of the power spectral density (PSD) of the naturalized river discharge over the one of the effective rainfall gives an experimental transfer function (ETF) (Pedretti et al., 2016; Schuite et al., 2019). The PSD of a signal is obtained by squaring the module of its Fourier transform, which is computed using a classical fast Fourier transform algorithm ('fft' function in MATLAB). An MCMC inversion procedure is implemented to adjust HYMIT to each ETF in order to estimate all controlling parameters for all sub-basins for which discharge data are available. The MATLAB package of Haario et al. (2006) for

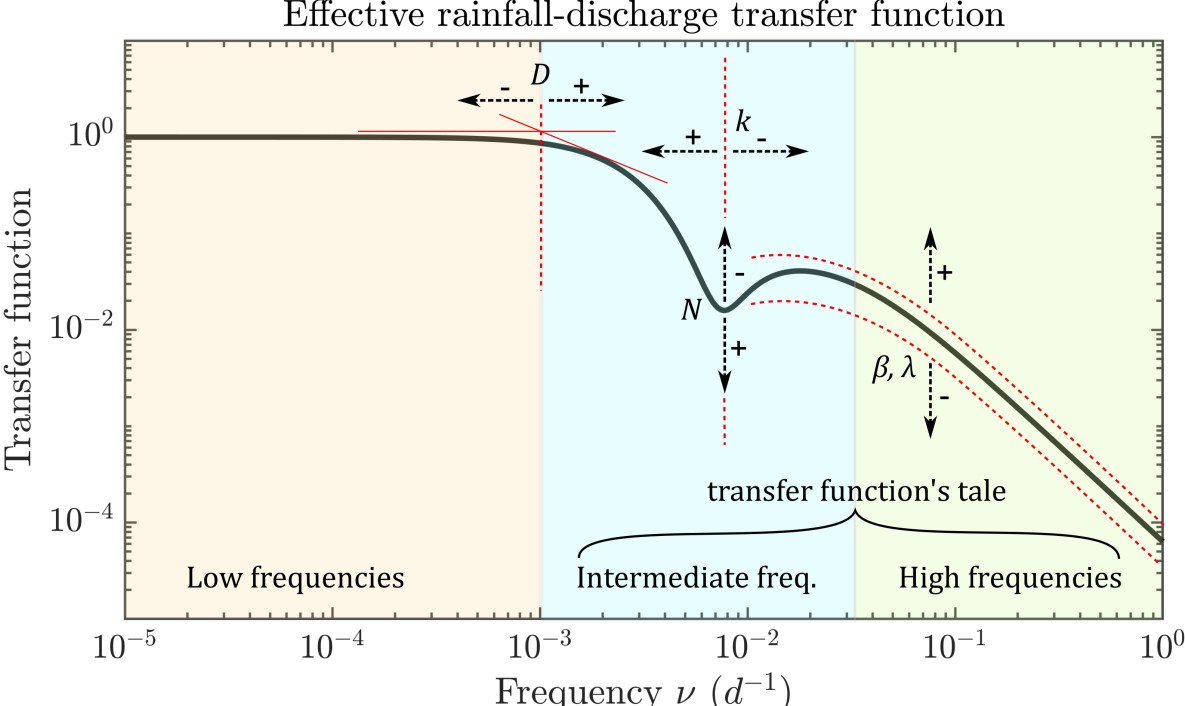

**Figure 5.** Example of effective rainfall to river discharge transfer function as modeled by HYMIT (Schuite et al., 2019), with a representation of the main influence of each parameter on its shape. The parameters are: $D$ the hydraulic diffusivity of the aquifer compartment, $N$ the number of reservoirs in the Nash cascade representing the flow transfer through the unsaturated zone and $k$ their emptying constant, $\beta$ the fraction of effective rainfall flowing through a watershed as surface runoff, and $\lambda$ the decay coefficient associated to it. Arrows accompanying a positive sign $+$ (respectively a negative sign $-$) show the TF's direction of change as a result of an increase (respectively a decrease) in the parameter's value. Note that the shape of HYMIT is highly variable and is ultimately the result of a complex interplay between all parameters which cannot be fully captured in this general depiction.

MCMC inversion is used with the built-in Metropolis–Hastings sampler. In the case of effective rainfall-discharge analysis with HYMIT, five parameters control the shape of the transfer function (TF), and thereby the climatic signal's transformation representation (Schuite et al., 2019) (Fig. 5):

- the hydraulic diffusivity of the aquifer compartment $D\,[m^2\,s^{-1}]$. This parameter controls the position of the first slope rupture in the TF, where it departs from a horizontal asymptotic line of equation $y = 1$. The rupture slides to higher frequencies for increasing values of $D$;

- the number of cascading linear reservoirs representing the transfer in the unsaturated layer $N\,[-]$;

– the emptying constant of these reservoirs $k\,[d]$. These two last parameters control the amplitude and partly the horizontal position of the local depression sometimes observed in the TF at intermediate frequencies. High values of $k \times N$ tend to exacerbate the prominence of this central energy loss;

     – the fraction of effective precipitation transiting to the outlet through the surface compartment $\beta\,[-]$. This parameter partly controls the shape of the transfer function at intermediate frequencies and high frequencies. Indeed, increasing $\beta$ gives

more energy (in terms of spectral density) to fast surface flow processes, thereby driving parts of the transfer function's tale upwards;

     – the characteristic runoff time scale $\lambda\,[d^{-1}]$. Together with $\beta$, this parameter modulates the shape and position of the TF's tale and may also affect the amplitude of the depression created by the transfer through the vadose zone.

Schuite et al. (2019) demonstrated that HYMIT is sensitive to all parameters on both a synthetic case study and real data,

but the shape of the TF is ultimately governed by a complex interplay between physical properties of hydrosystems and the characteristic flow time scales they induce in each hydrologic compartment. For instance, the central energy depression in the TF, known to appear in the presence of a strongly inertial unsaturated zone, was observed for the Essonne watershed but not for the Aube watershed, yet both nested within the Seine basin (Schuite et al., 2019).

The method requires knowledge of the effective rainfall, which is not directly measured. It therefore needs to be assessed

by a surface balance model. In this study, the surface module of the hydrological model CaWaQS3.02 is used (see section 2.2) both for practical reasons (spatialized soil, land surface data availability, and integration) and for consistency concerns with subsequent fitting steps. The fitting of the surface balance model is detailed in the next section 2.4.

A total of 384 discharge gauging stations are (or were) operational across the different rivers of the Seine basin. Daily discharge data were checked for completeness and overall quality. In the presence of any gap longer than 10% of the total

series length, the station was discarded from the analysis. So were stations with overly short records (less than 10 years), to maximize the statistical power of the analysis. Small gaps are filled by linear interpolation. Minor data errors were corrected, such as inconsistent timelines, double records, or negative values. After data curation and selection, discharge series from 221 stations remained. Prior to Fourier transformation, all time series are detrended and windowed using a Hanning tapering function.

**2.4   Step-wise fitting methodology based on a nested hydrological approach**

A nested fitting method is proposed (Fig. 6). It is based on the identification of the CaWaQS3.02 model parameters, process by process, on the basis of measurable quantities. It is considered nested due to the conditionality of the parameter fields on their forcings. Each step is developed with this conditionality idea in mind. Roughly, the first step considers the estimation of the partitioning of the rainfall into AET and effective rainfall, while the next one focuses on the partitioning of effective rainfall

into fast runoff and slow infiltration. The last steps deal with regulating the velocities of both surface and subsurface flows.

The first step of the fitting methodology consists in estimating the total AET at the basin scale from the average discharge of water flowing out of the basin (Fig. 6, step 1). In order for this average to exist mathematically, it is then necessary to

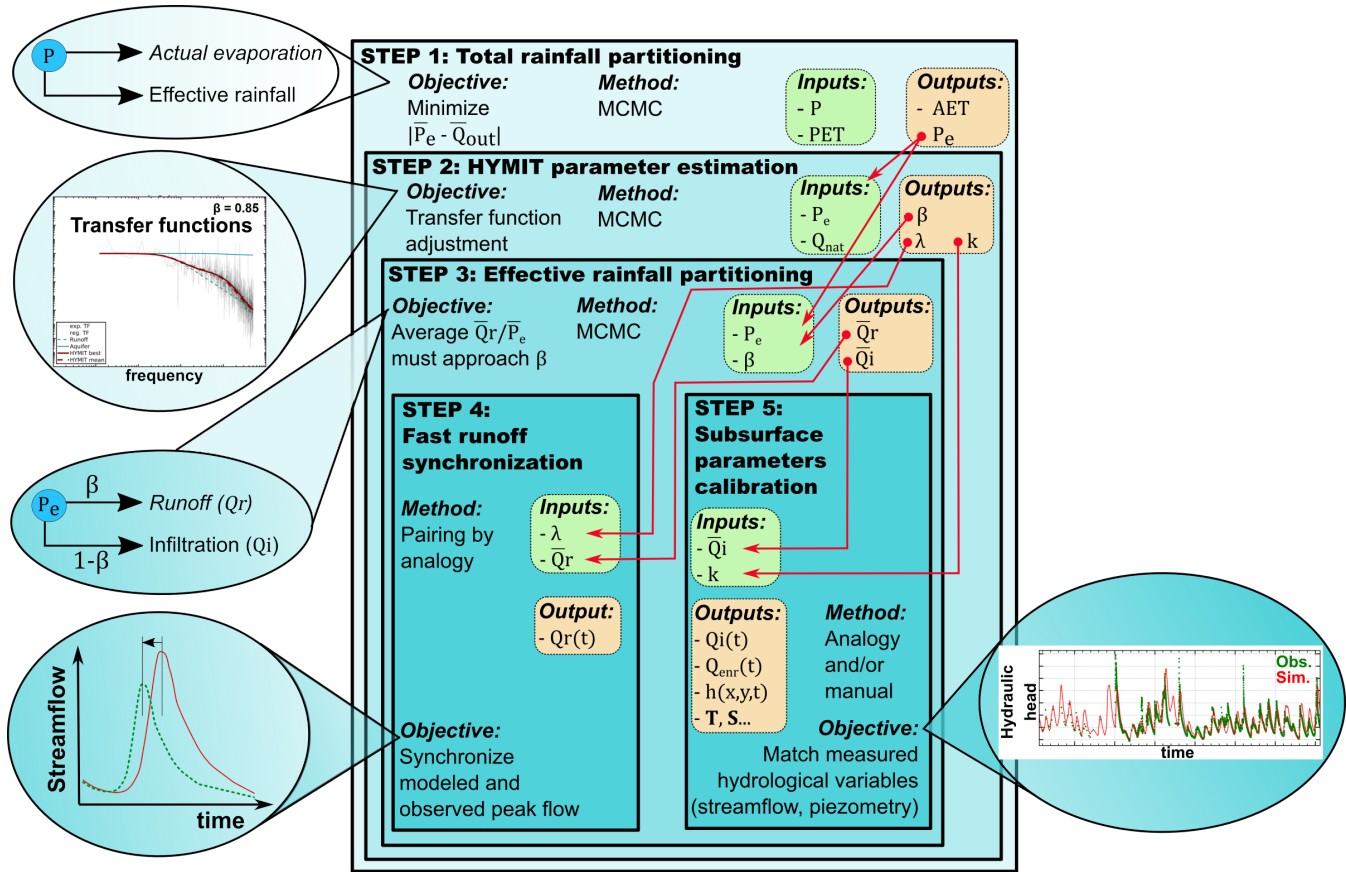

**Figure 6.** Illustrated flow chart of the full step-wise HYMIT–CaWaQS fitting procedure. Notations with overbars indicate integrated variables over time. Variable space and/or time dependencies are mentioned using $X(x,y,t)$-type notations: $AET$: actual evapotranspiration, $\beta$: effective precipitation fraction transiting to the catchment outlet through the surface compartment, $h$: hydraulic head, $k$: emptying constant of Nash-cascade linear reservoir, $\lambda$: characteristic runoff time scale, $P$: rainfall, $P_e$: effective rainfall, $PET$: potential evapotranspiration, $Q_{enr}$: stream–aquifer exchange flow, $Q_i$: infiltration flow, $Q_{nat}$: naturalized observed river discharge, $Q_{out}$: discharge at sub-basin outlet, $Q_r$: runoff flow, $T$: transmissivity coefficient and $S$: storage coefficient.

reproduce the flows averaged over 17 years, which is the stationarity period of this signal (Flipo et al., 2012; Massei et al., 2010, 2017). In the surface balance module, the set of parameters regulating actual evapotranspiration (AET) flow simulation are adjusted using an MCMC approach aiming at minimizing the discrepancy between the long-term average discharge at the watershed outlets and the long-term average meteoritic net input over the watershed impluvium (*i.e.*, the effective rainfall that corresponds to water available for flow within the basin). Furthermore, we selected a total of 35 reference sub-basins across the Seine hydrosystem satisfying two criteria, namely, good spatial coverage of the basin as well as long and complete daily discharge time series.


Before proceeding with the fitting of the model parameters themselves, sets of distributed minimalist hydrological parameters are estimated at 221 gauging stations in the basin using the HYMIT method (Fig. 6, step 2). The following steps of the fitting procedure are based on these estimates with the objective each time to spatially reproduce the variability of these minimalist parameters with the process-based model used, *e.g.*, CaWaQS. In other words, the wealth of information provided by the quantified and regionalized estimates of HYMIT parameters is used as the only support for the fitting of the surface and subsurface water flow calculation within the model. Thus, this step enables the transition from a point assessment to a continuous spatio-temporal characterization of the regional hydrosystem behavior.

The third step consists in adjusting a parameter of the water production cells in order to reproduce as closely as possible the effective rainfall partitioning evaluated by HYMIT ($\beta$ coefficient, Fig. 6, step 3).

The fourth step deals with the temporal synchronization of fast runoff estimated by the process-based model using the estimate of this quantity obtained with HYMIT at each discharge gauging station ($\lambda$ coefficient, Fig. 6, step 4). The attribution of $\lambda$ coefficient values identified with HYMIT to the corresponding CaWaQS parameter is made by analogy. For each gauging station, a spatial analysis is performed to calculate the proportion of the CaWaQS hydrological units that comprise the catchment of the station. Over the Seine basin, it is possible to identify a sufficient number of upstream catchments that are mostly composed of a single CaWaQS hydrological unit. For each of these catchments, the CaWaQS parameter is set up in a straightforward way, simply equating it with the value of the HYMIT $\lambda$ coefficient.

Until now, all the steps were focused on surface processes. The proper simulation of these processes is crucial since they allow us, among other processes, to estimate a distributed recharge toward the aquifer system. The fifth and last step of the methodology is carried out differently for the vadose zone than for the aquifer system and is divided into three substeps :

- a preliminary coarse calibration of transmissivity parameter fields for each model aquifer layer, performed in steady state. A trial-and-error approach is used to adjust simulated water tables to mean level values measured at control wells. To achieve this step, an aquifer compartment-only simulation is implemented. A mean aquifer recharge field, calculated over two 17-year cycles (*i.e.*, 1986–2020 period), derived from the HYMIT-calibrated CaWaQS surface module, is used to constrain the simulation ($\overline{Q_i}$ on Fig. 6, step 5);

- a fitting of the Nash-cascade parameters, namely, $k$ and $N$ (see sections 2.3 and 2.2). Usually associated with lithology, HYMIT-$k$ values are spatially distributed along a functional sectorization combining two information types: dominant soil textures (Fig. 1) and upmost free aquifer formation lithology (Fig. 3). For each association identified, attribution of $k$ values is made using an analogy method, as previously described in the case of HYMIT-$\lambda$ coefficients distribution (see section 2.4). $N$ parameter field, reflecting unsaturated zone thickness, is geometrically defined, based on the difference between subsurface adjusted water levels in steady state and ground elevation, while considering a single reservoir to be representative of an elementary 5-m thickness. Overall, at the basin scale, $k$ parameter values range from 2 to 9 days. Tertiary terrains are mostly associated with low thickness values, from 0 to 20 $m$, while ranging up to 40 $m$ within the Chalk impluvium limits (Fig. 12). An unsaturated thickness map (not shown) noticeably depicts high values, up to 135 $m$, in the southern part of the east border of the basin;

– finer manual adjustments of parameter fields regulating both water levels and dynamics, namely, transmissivity ($T$),

storage ($S$), and conductance ($C$) coefficients. To do so, iterative transient state coupled model runs are performed, constrained by daily time-step HYMIT-calibrated infiltration fluxes ($Q_i(t)$ in Fig. 6, step 5). $T$ and $S$ parameter distributions are manually tuned, based on expert knowledge of regional aquifer functioning, and also accounting for piezometric reference maps as well as pumping test values, where available. In order to minimize the number of parameters to be adjusted, conductance fields, which regulate surface–subsurface interactions (*i.e.*, stream–aquifer exchanges and overflows from aquifer to surface) are automatically calculated and updated consistent with successive $T$ fields following the methodology proposed by Rushton (2007), and implemented in a previous version of the Seine model by Pryet et al. (2015), and also successfully used on the Loire basin by Baratelli et al. (2016). The performance of each trial is assessed at the scale of each control point, combining statistical criteria calculations and visual comparisons between simulated and observed time series (see sections 3.2 and 3.3). This initiates a trial-and-error fitting process, alternating between hydrodynamic parameter modifications and run quality evaluation, until satisfactory performance is reached.

Prior to any subsurface fitting work, GW withdrawals are integrated into the application in order to account for anthropogenic-induced disturbances on the aquifer system. Data consist in spatially located annual volume time series. Depending on the type of water usage, annual values were either linearly distributed over the year (drinking water, industry) or specifically dispatched over the summer season (irrigation).

The next two sections exemplify the power of the step-wise methodology applied to the regional scale of the Seine basin.

## 3 Results: Performance of the coupled model

As mentioned in section 2, the step-wise fitting methodology relies on a distributed assessment of physical parameters that control the partitioning of hydrological fluxes within watersheds. Before reviewing the performance of the coupled CaWaQS–Seine model, we therefore qualitatively review the results of the distributed HYMIT analysis at the Seine basin scale. The subsections 3.3 and 3.2 propose a more classic assessment of the model performance, comparing simulated values with measurements of river discharges and GW hydraulic heads.

### 3.1 Qualitative analysis of spatially distributed infiltration fluxes estimated with HYMIT

The average partitioning of effective rainfall between surface runoff and deep infiltration is adjusted based on the HYMIT $\beta$ parameter. This parameter is generally poorly constrained at large scale, especially its spatial distribution. Hence, very little information is available to compare our $\beta$ estimates against, and ultimately, to validate them.

Fortunately, the French Geological Survey (BGRM) has developed a systematic method to qualify the propensity of terrains to either infiltrate water or to generate runoff, as described by Mardhel et al. (2021). The method takes advantage of the mismatch evaluated between thalweg locations, inferred from digital elevation models, and the actual location of drainage networks, used to build a normalized index called IDPR (Network Development and Persistence Index). The lower its value, the more a terrain is prone to infiltration and *vice versa*. Mardhel et al. (2021) calculated this index for the entire French

metropolitan area at high resolution (25 $m$). We take this opportunity to qualitatively assess the consistency of $\beta$ estimates, which, in a way, is very complementary to the IDPR index, being much less spatially resolved but providing an actual operable value to the flow partitioning, which is of primary importance for modeling purposes.

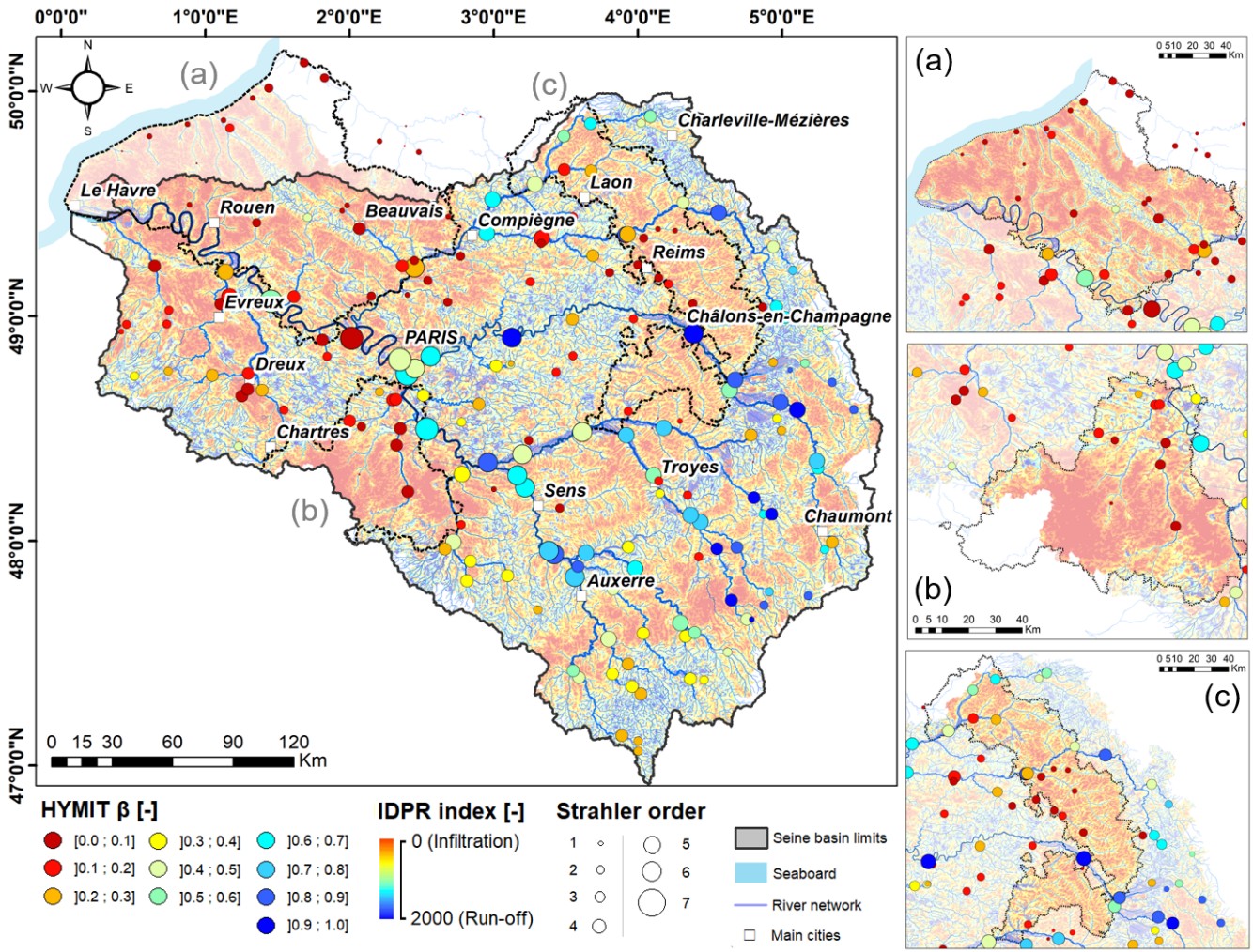

**Figure 7.** Spatial comparison between HYMIT $\beta$ and IDPR values at the Seine basin scale. Local focuses : (*a*) "Pays de Bray / Pays de Caux", (*b*) "Beauce" and (*c*) "Craie Champenoise."

Figure 7 superimposes estimates of $\beta$ at each discharge gauging station (dots) and the IDPR map produced by Mardhel et al.
(2021). First, an overall spatial coherence is observed between these two indicators. Upstream stations experience medium-to-high runoff components ($\beta > 0.2$), especially in the eastern and southern fringes of the basin. These regions entail a dominant proportion of medium-to-high IDPR areas as well. Conversely, areas dominated by low IDPR values are drained by rivers where low values of $\beta$ are found. We further note a satisfactory sectorial coherence in $\beta$ estimates: Two low-order rivers

draining the same geomorphological units exhibit close properties in terms of flow partitioning. This aspect is particularly well established in the "Beauce" and "Craie Champenoise" regions (Fig. 7, panels (b) and (c), respectively). It is also the case for the southern regions of the Morvan (see Fig. 2); yet in this case $\beta$ values seem comparatively low in regard to the high IDPR values, possibly underscoring a more complex interplay between different runoff-generating mechanisms in this sector (slope, soil structure, geology, etc.).

Another marker of consistency is present in the northwestern part of the basin, in and around the "Pays de Bray" sector (panel (a) in Fig. 7). The "Pays de Bray" anticline is characterized by the local emergence of sandy and clayey Jurassic terrains with high slopes, extending NW–SE (see Fig. 2), among Upper Cretaceous carbonate subplanar units (see Fig. 4). Therefore, property contrasts between these terrains are clearly distinguishable in IDPR values and are also well captured by the flow partitioning estimation with the HYMIT, as the river draining the "Pays de Bray" anticline displays a $\beta$ value of approximately 45 % to be compared with the very low values found on adjacent stream networks ($\beta < 10$ %).

## 3.2 Simulation of hydraulic heads in aquifers

Within the modeled aquifer system limits, 340 head observation time series are compiled from the national ADES database (https://ades.eaufrance.fr/). Raw data curation is performed based on the minimum covered time-span and a threshold number of observation data. Time series on which GW dynamics are not clearly identifiable are discarded as well as measurement sites located in local non-modeled formations (*e.g.*, perched water tables). In all, 269 GW control points with data suited for aquifer calibration are selected.

Calculated over the 2003–2020 calibration period, $RMSE$ (Root Mean Square Error), $MAE$ (Mean Absolute Bias), as well as Pearson correlation coefficient and $KGE$ (Kling–Gupta Efficiency) (Gupta et al., 2009; Kling et al., 2012) are used to assess model performance (Tab. 1). GW simulation exhibits satisfactory performance overall as nearly two thirds of control points are associated with both $RMSE$ and $MAE$ values below 4 $m$ (63 % and 68 %, respectively). A total of 66 % of piezometers show correlation coefficients above 0.5, demonstrating the model ability to mimic the great multiplicity of aquifer dynamics encountered at the regional scale. Less clear-cut performance results (43 % of points) are obtained regarding a proper joint reproduction of the evolution of both levels and dynamics ($KGE > 0.5$). General aquifer $RMSEs$ and $MAEs$ are 5.4 and 4.7 $m$, respectively. At the model layer scale, mean $RMSE$ and $MAE$ fit in the 0.9–6.7 $m$ and 0.4–6.5 $m$ ranges, respectively. Unsurprisingly lower values are calculated for alluvial formations (1.4 $m$, 1.2 $m$) and the Jurassic ensemble (1.5 $m$, 0.6 $m$), as most control points are constrained by proximal river drainage levels. $RMSE$ scores map (Fig. 8) shows a homogeneous distribution of lower values over the entire basin, apart from the Evreux–Dreux area (left bank of the downstream Seine River, see Fig. 2) gathering the highest misfits, where local disturbances of aquifer flows are known to be heavily karst-induced (El Janyani et al., 2012), making them harder to reproduce by the model.

**Table 1.** Distribution of coupled model performance criteria (2003–2020 period) on GW simulation. Upper table (range values in $m$) : Root Mean Square Error ($RMSE$), Mean Absolute Error ($MAE$). Lower table (non-dimensional range values): Pearson correlation coefficient ($C_{pearson}$) and Kling–Gupta Efficiency coefficient ($KGE$).

| | $RMSE$ | $MAE$ | $RMSE$ | $MAE$ |
|---|---|---|---|---|
| Value range $[m]$ | Piezometer count [-] | | Cumulative percentage [%] | |
| [0.0–2.0[ | 94 | 131 | 34.9 | 48.7 |
| [2.0–4.0[ | 75 | 51 | 62.8 | 67.7 |
| [4.0–6.0[ | 32 | 29 | 74.7 | 78.4 |
| [6.0–8.0[ | 23 | 18 | 83.3 | 85.1 |
| [8.0–10.0[ | 14 | 13 | 88.5 | 90.0 |
| > 10.0 | 31 | 27 | 100.0 | 100.0 |
| | $C_{pearson}$ | $KGE$ | $C_{pearson}$ | $KGE$ |
| Value range $[-]$ | Piezometer count [-] | | Cumulative percentage [%] | |
| ]0.7;1.0] | 100 | 35 | 37.2 | 13.0 |
| ]0.5;0.7] | 78 | 59 | 66.2 | 35.0 |
| ]0.4;0.5] | 24 | 21 | 75.1 | 42.8 |
| ]0.2;0.4] | 34 | 35 | 87.7 | 55.8 |
| ]0.0;0.2] | 17 | 23 | 94.0 | 64.3 |
| ]-∞;0.0] | 16 | 96 | 100.0 | 100.0 |

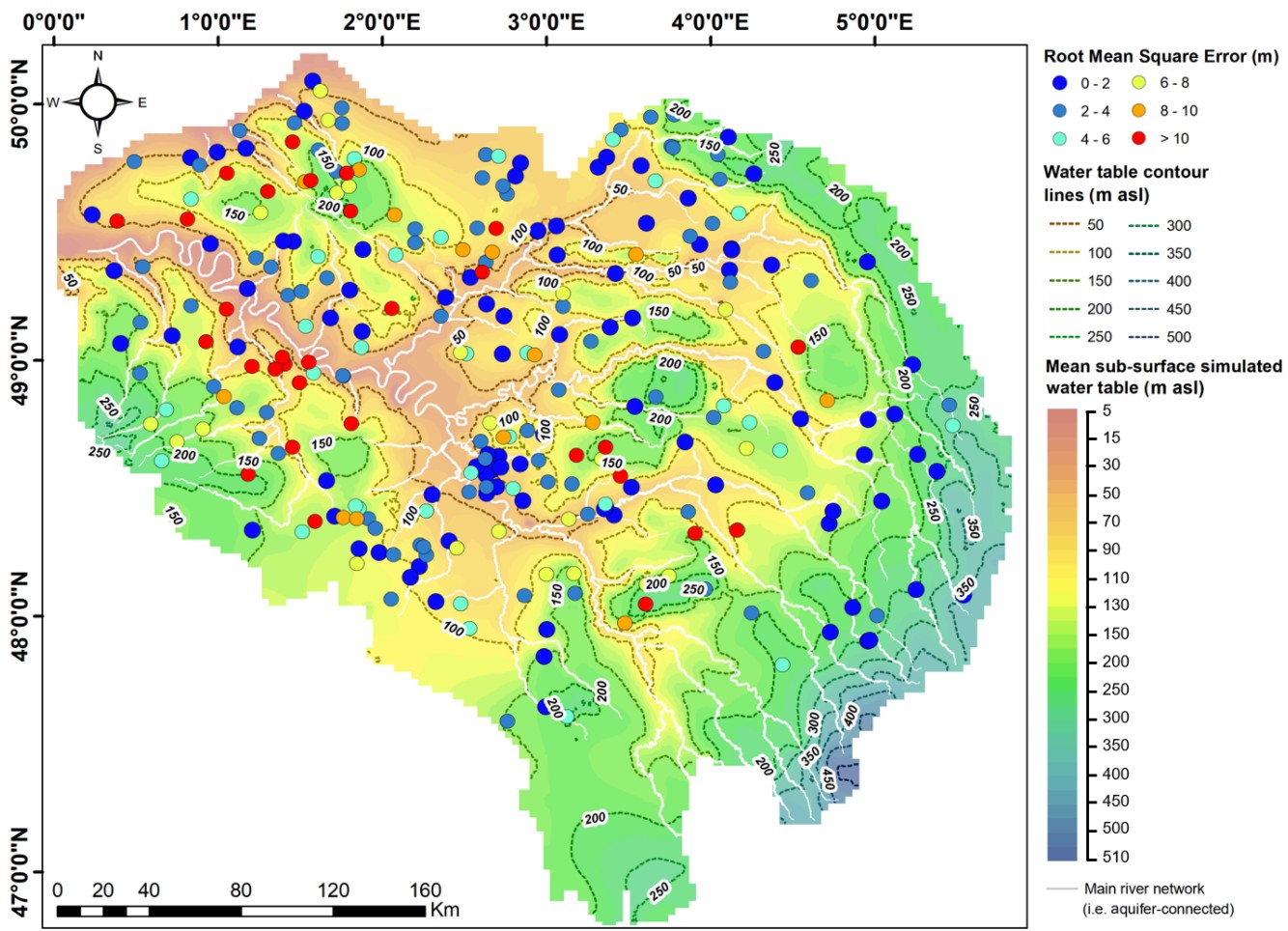

**Figure 8.** RMSE criteria for GW simulation over the 2003–2020 period (see also Table 1). Water table contour lines (50-m span) and background map depict spatial evolution of simulated mean GW levels for all uppermost aquifers (subsurface) over the same period. *asl* : Above Sea Level.

### 3.3 Simulation of river discharges

Similarly to raw piezometry data, a pre-processing work is carried out on river discharge observations using data compiled from the HYDRO database (http://hydro.eaufrance.fr/), totalling 384 river stations (see section 2.3). As previously stated, since calculations of stream–aquifer exchanges are constrained to the main river network (see section 2.2 and Fig. 4), 167 discharge gauging stations only are considered as valid discharge calibration control points. Usual Nash–Sutcliffe (Nash and Sutcliffe, 1970) and Kling–Gupta efficiency coefficients (labeled $NSE$ and $KGE$, respectively, in Tab. 2) are selected to evaluate simulation performances on river discharges.

At the basin scale, 50 and 53 % of control stations, respectively, show $NSE$ and $KGE$ scores above or equal to 0.5. The spatial distributions of both criteria (not shown) demonstrate that most of the higher values are noticeably distributed along the Seine River and its eight main tributaries (see bold blue lines in Fig. 2). For the 64 stations located along this network portion (2,910 $km$, 43 % of total modeled network), 54 (resp. 26) of them are associated with KGE $\geq$ 0.5 (resp. 0.7). Noticeably, 44 stations combine both $NSE$ and $KGE$ values $\geq$ 0.5.

**Table 2.** Distribution of coupled model performance criteria (2003–2020 period) on river discharge simulation. Non-dimensional range values. Nash–Sutcliffe efficiency ($NSE$) and Kling–Gupta efficiency coefficients ($KGE$).

| | $NSE$ | $KGE$ | $NSE$ | $KGE$ |
|---|---|---|---|---|
| Value range [−] | Gauging station count [−] | | Cumulative percentage [%] | |
| ]0.7;1.0] | 38 | 41 | 22.8 | 24.6 |
| ]0.5;0.7] | 45 | 49 | 49.7 | 53.9 |
| ]0.4;0.5] | 20 | 12 | 61.7 | 61.1 |
| ]0.2;0.4] | 23 | 40 | 75.4 | 85.0 |
| ]0.0;0.2] | 13 | 8 | 83.2 | 89.8 |
| ]-∞;0.0] | 28 | 17 | 100.0 | 100.0 |

## 4 Analysis of hydrological fluxes within the regional Seine basin

Once the CaWaQS–Seine model exhibits satisfactory performances (see section 3), it is used to estimate spatio-temporally distributed key hydrological fluxes such as effective rainfall (see subsection 4.1), infiltration rates (see subsection 4.2), surface runoff, as well as exchanges between aquifer units or between aquifers and rivers (see subsection 4.3). Finally, all these data are synthesized by the Seine basin water balance, including water fluxes between aquifer units (see subsection 4.4).

### 4.1 Distribution of effective rainfall

CaWaQS–Seine is first used to estimate the distribution of two important hydrological quantities over a period of 17 years (2003–2020): AET and effective rainfall (Fig. 9$c$ and Fig. 9$d$ respectively), the latter considered to be stationary during this period (see subsection 2.4). They are both estimated from rainfall and PET (Fig. 9$a$ and Fig. 9$b$).

The effective rainfall is highly contrasted across the Seine basin, ranging from 44 $mm\,a^{-1}$ up to 918 $mm\,a^{-1}$ locally (Fig. 9$d$). The lowest effective rainfall rates occur around the city of Paris and in the Eure basin, around Chartres and Dreux (see Fig. 2). The central part of the basin experiences an effective rainfall mainly lower than 130 $mm\,a^{-1}$, while the eastern ridge, or Jurassic edge, experiences effective rainfall rates higher than 250 $mm\,a^{-1}$, sometimes reaching 920 $mm\,a^{-1}$ as is the case in the southern part of the basin in the Morvan area. It is also the case in Normandy, in the northern part of the basin, called "Pays de Caux" (see Fig. 7, panel ($a$)), in the north of Rouen.

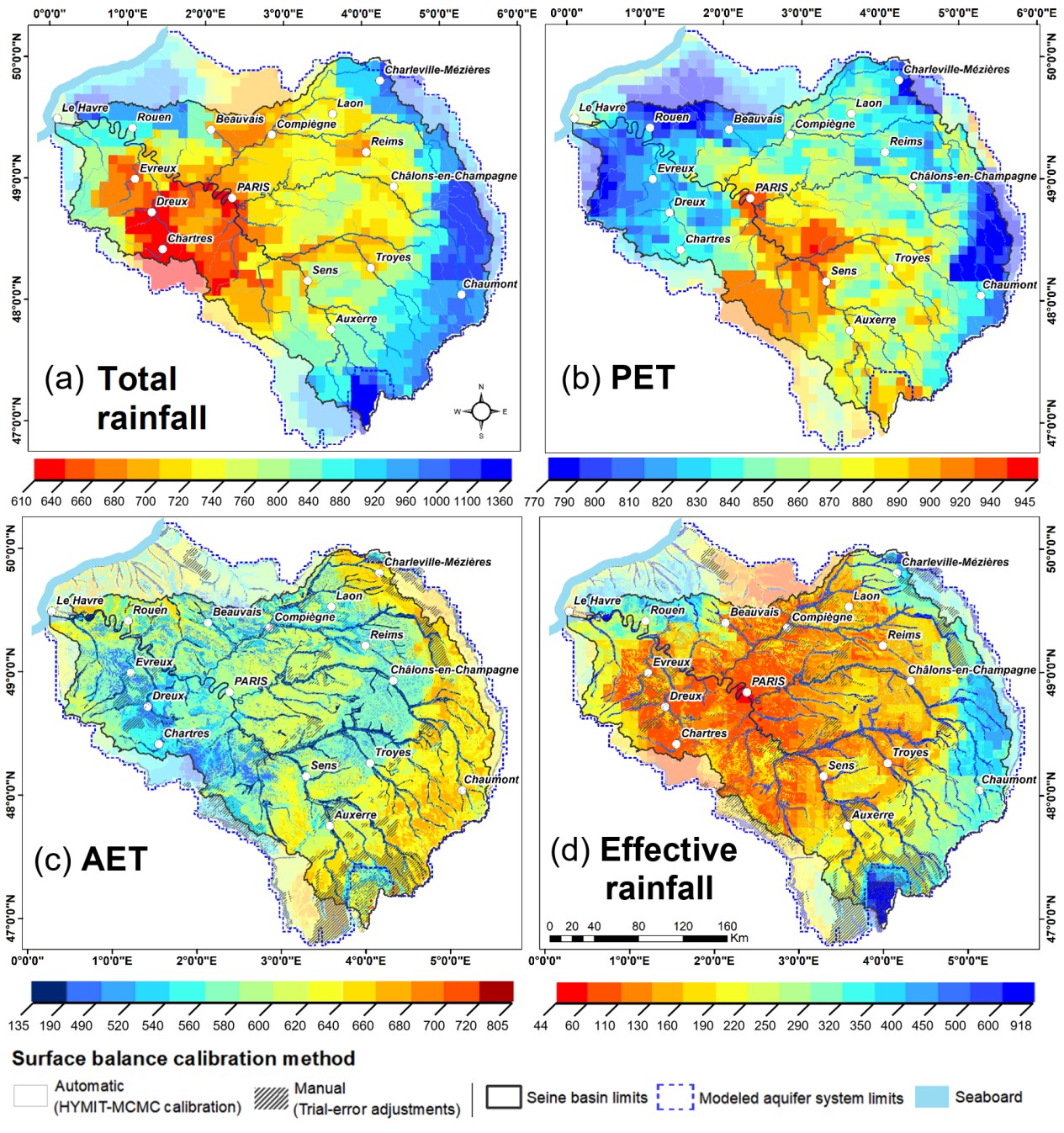

**Figure 9.** Mean distribution fields of (*a*) 'total rainfall' (*i.e.* sum of liquid rainfall and snowfall, which is almost nil at the Seine basin scale), (*b*) PET, (*c*) AET, and (*d*) effective rainfall, over the 2003–2020 period, in $mm\,a^{-1}$.

## 4.2 Distribution of infiltration rates

A key piece of information for hydrogeologists and groundwater managers is the estimation of aquifer recharge. A complete water balance calculation using the CaWaQS model enables the simulation of effective rainfall, runoff, and infiltrated water distributions at the daily time-step with a high spatial resolution. It is therefore possible to represent the average distributed partitioning of effective rainfall at the basin scale (Fig. 10), over the period 1970–2018, expressed as the ratio between the simulated runoff and the total effective rainfall.

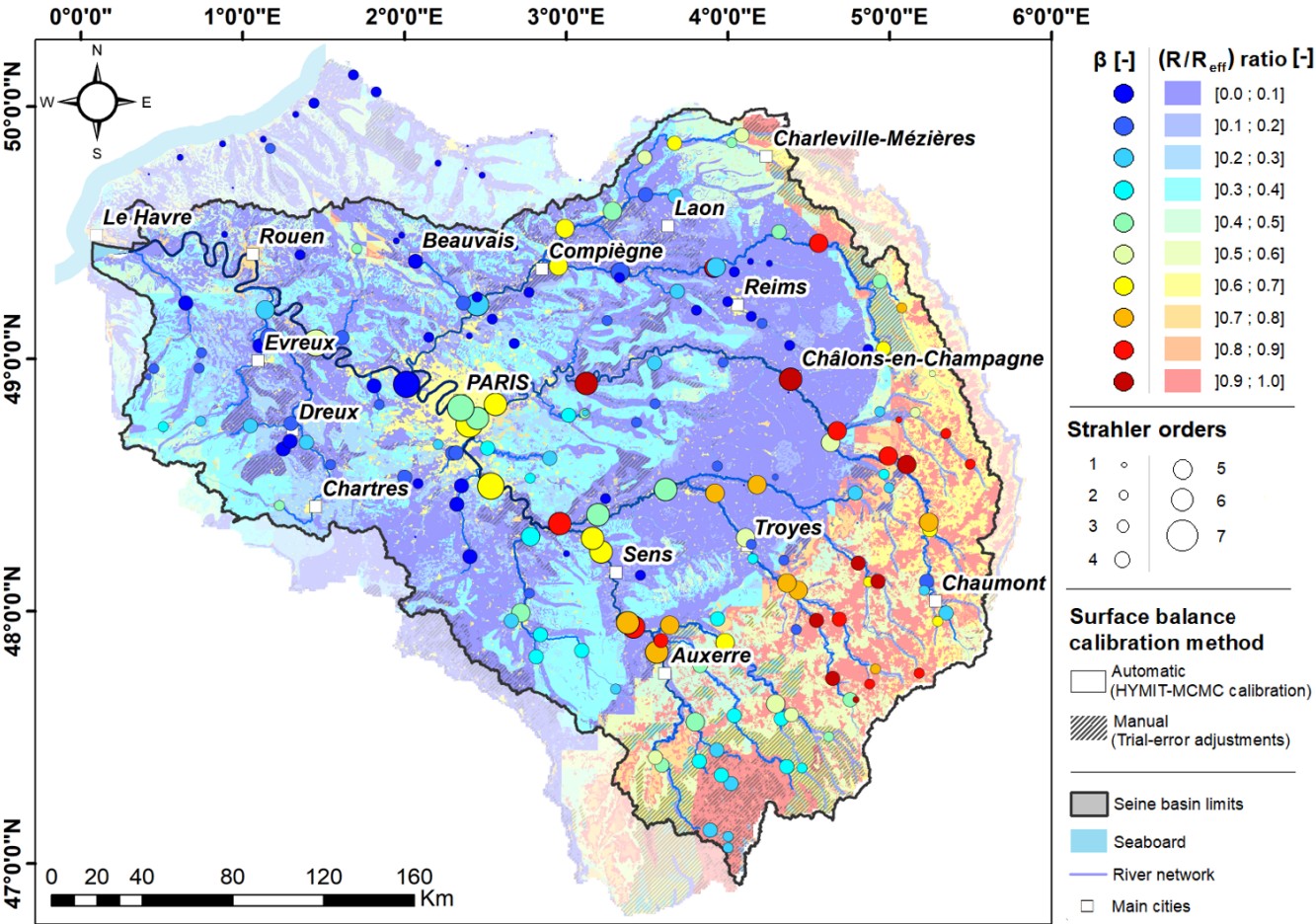

**Figure 10.** Distributed partitioning of the pluri-annual mean effective rainfall, calculated over the 1970–2018 period, and expressed as a ratio between simulated runoff fraction ($R$) and total effective rainfall ($R_{eff}$). HYMIT $\beta$ values determined at discharge gauging stations are also indicated using point symbols.

Before analyzing further the estimated infiltration rates, it is important to note that the distributed effective rainfall partitioning estimated by CaWaQS–Seine is consistent with that obtained using the HYMIT analysis alone (Fig. 10), which is in

agreement with the most advanced GIS-based analysis (section 3.1). Indeed, the simulated predominance of the runoff process
over the entire Jurassic edge of the basin (simulated local partitioning generally higher than 0.6) is in agreement with high $\beta$
values in this area. The opposite observation can be made for the interior of the basin with low simulated partitioning values
that agree with HYMIT low partitioning values for head water streams in the area (lower than 0.3, Fig. 10).

Distributed infiltration rates are then calculated, as an inter-annual average over the simulation period 1970–2018 (Fig. 11).
They balance the effective rainfall mostly leading to relatively moderate infiltration rates (lower than 120 $mm\,a^{-1}$) over the
basin, except **(i)** on the Normand "Pays de Caux" north of Rouen where it can reach 400 $mm\,a^{-1}$, and **(ii)** to a lesser extent
over the Chalk area. In particular, infiltration rates are controlled by the geology as it is lower in areas with a high effective
rainfall than in areas with lower effective rainfall, 123 $mm\,a^{-1}$ over the Jurassic and lower Cretaceous aquifer units and 151
$mm\,a^{-1}$ over Upper Cretaceous and Tertiary aquifer units (Tab. 3).

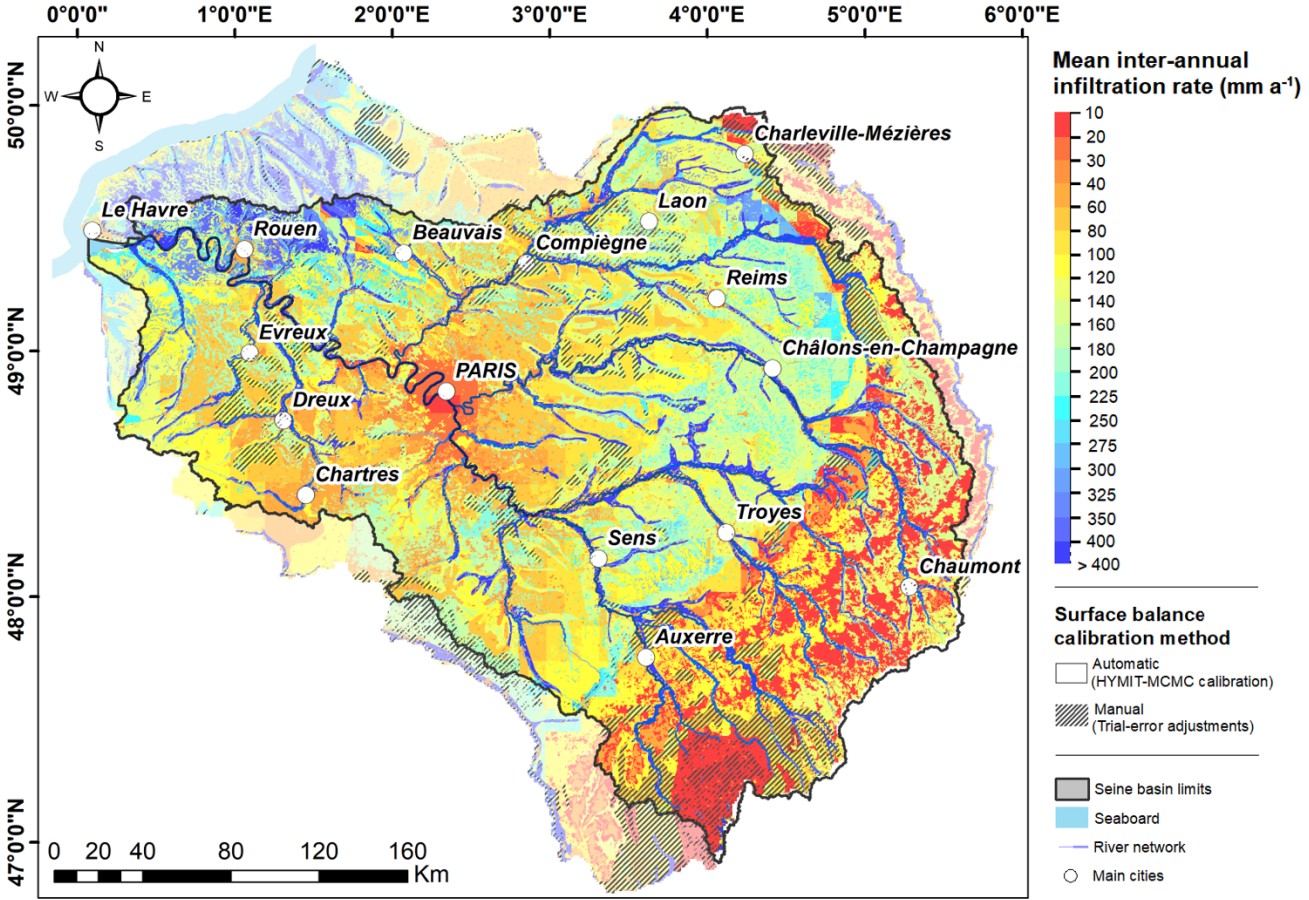

**Figure 11.** Mean annual infiltration rates, in $mm\,a^{-1}$, simulated by CaWaQS–Seine. Mean values over the 1971–2019 period.

There is also a spatial coherence between $\beta$ and the infiltration rate from the point of view of land use, insofar as the
predominantly agricultural (resp. urban) areas are indeed marked by higher (resp. lower) infiltration (see, for instance, the
Chalk area or the central Île-de-France that surrounds Paris city). Also, the areas initially associated with very low IDPR
values (fast infiltrating areas – see Fig. 1) clearly appear on the infiltration map, with infiltration rates higher than the regional
average (see, for instance, the Pays de Caux area or the eastern of Chartres).

Finally, geological characteristics also emerge from the zoning, insofar as the sectors dominated by soils classically more
permeable and conductive to infiltration (e.g., sands, alluvium) delimit areas of higher infiltration rates than the regional trend.
Such configuration is noticeable for regions such as the Sologne or Fontainebleau forest areas (see Fig. 1), or Thanetian sands
and alluvium deposits (see Fig. 4).

## 4.3 Distribution of groundwater contribution to river discharges

Labarthe et al. (2015) and Pryet et al. (2015) were the first to publish a spatially distributed evaluation of stream–aquifer
exchanges at the regional scale, exemplified with the Seine basin. At the time, they followed the step-wise fitting methodology
of Flipo et al. (2012). Given the advances proposed in the current paper, we re-assessed those estimates (Fig. 12) based on
the HYMIT functional analysis of the hydrological behavior of many sub-basins and also on the extension of the subsurface
domain that is taken into account by the model.

The pluri-annual average (2003–2020) contribution of groundwater (GW) to river discharge is calculated along the river
network of the Seine basin. Two specific patterns of spatial GW contribution to river baseflow appear in the Seine basin:

– a longitudinal increase in the contribution of GW to river baseflow from upstream to downstream for river systems
originating in the Jurassic edge of the basin;

– a very high contribution of GW to river baseflow over the tertiary ($> 0.75$), alluvial ($> 0.4$), or Upper Cretaceous (mostly
$> 0.6$) aquifer units (Fig. 12, panel ($b$)).

Those patterns are confirmed by a spatial analysis of the statistical distribution of GW contribution to river baseflow regarding
the specific discharge. Each analysis is made for each outcropping aquifer unit. For the Jurassic and Lower Cretaceous aquifer
units, the lower the specific discharge, the higher the GW contribution to river baseflow with a highest limit at 0.7 for a few
small streams in the area (Fig. 12, panel ($b_4$)). For the small streams fed by GW from the Tertiary aquifer units, the GW
contribution is always higher than 0.8 (Fig. 12, panel ($b_2$)). The GW contribution of the Upper Cretaceous aquifer unit is
similar to that of the Tertiary aquifer units (Fig. 12, panel ($b_3$)), even though the absolute value of the contribution can be
slightly lower in this area (0.5). As for the Jurassic and Lower Cretaceous edge of the basin, the GW contribution of alluvial
GW to river discharge exhibits a decreasing GW contribution with the increase of the specific discharge that is correlated to
higher Strahler orders (Fig. 12, panel ($b_1$)). But the GW contribution in alluvial aquifers is always higher than 0.4, which is a
significant difference with the Jurassic edge.

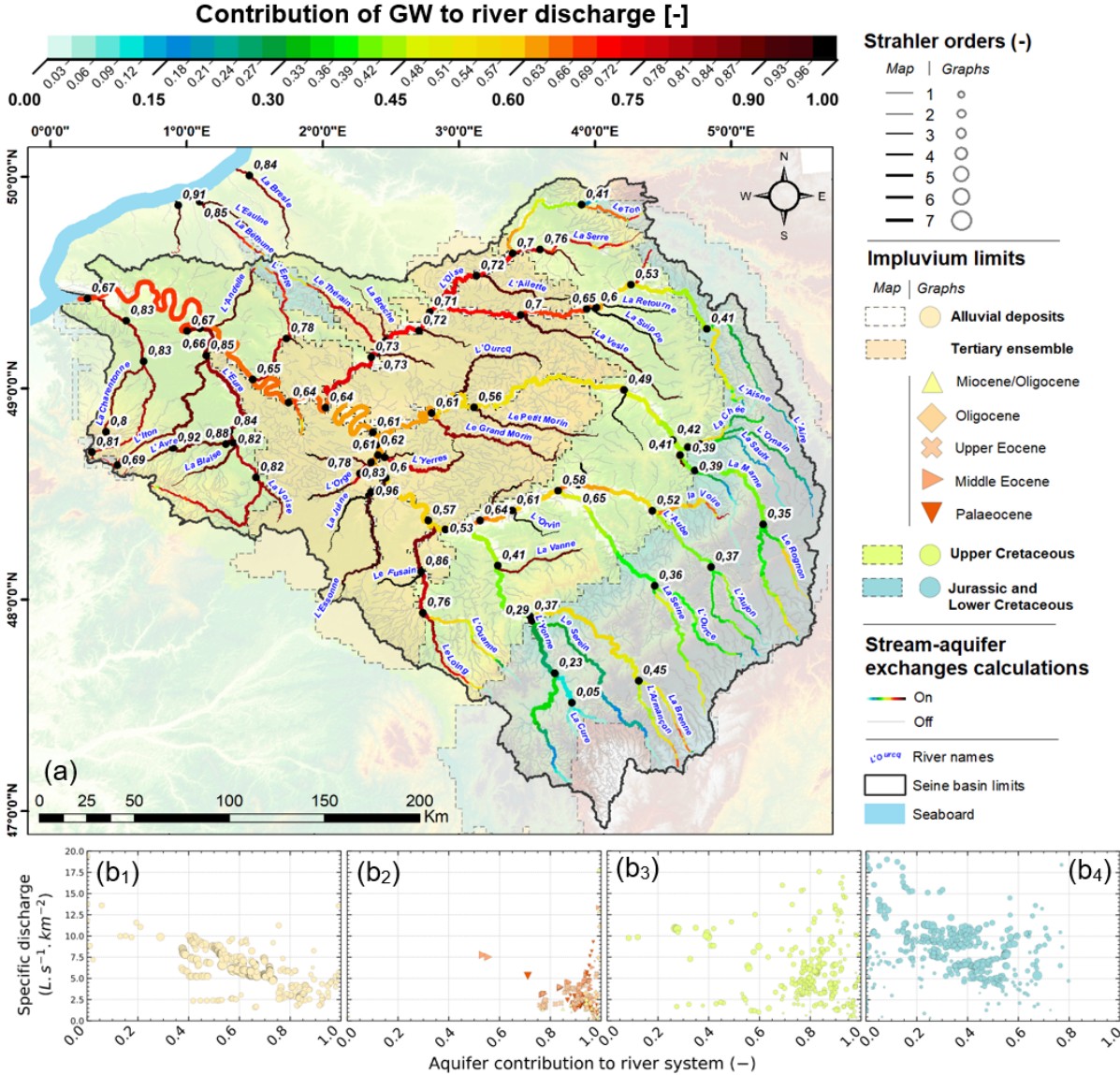

**Figure 12.** (*a*) Distribution, at the river reach scale, of the aquifer system contribution to the river network. For each reach, values are expressed as a fraction ($\in [0;1]$) of the network total input volume, calculated within the limits of its respective upstream watershed. Mean annual values calculated over the 2003–2020 period. For the sake of readability, labels indicate the fraction value right after each confluence and outlets of the main network. (*b*) Relations between specific discharges in river (in $L\,s^{-1}\,km^{-2}$) and aquifer system contribution ($-$) to hydraulic network. Each point corresponds to a river reach. Subplots gather data according to the layers the network is connected to. From most recent (far left) to oldest (far right): ($b_1$) alluvial deposits, ($b_2$) Tertiary ensemble, ($b_3$) Upper Cretaceous/regional Chalk aquifer and ($b_4$) Jurassic and Lower Cretaceous ensemble.

 ## 4.4  Water budget

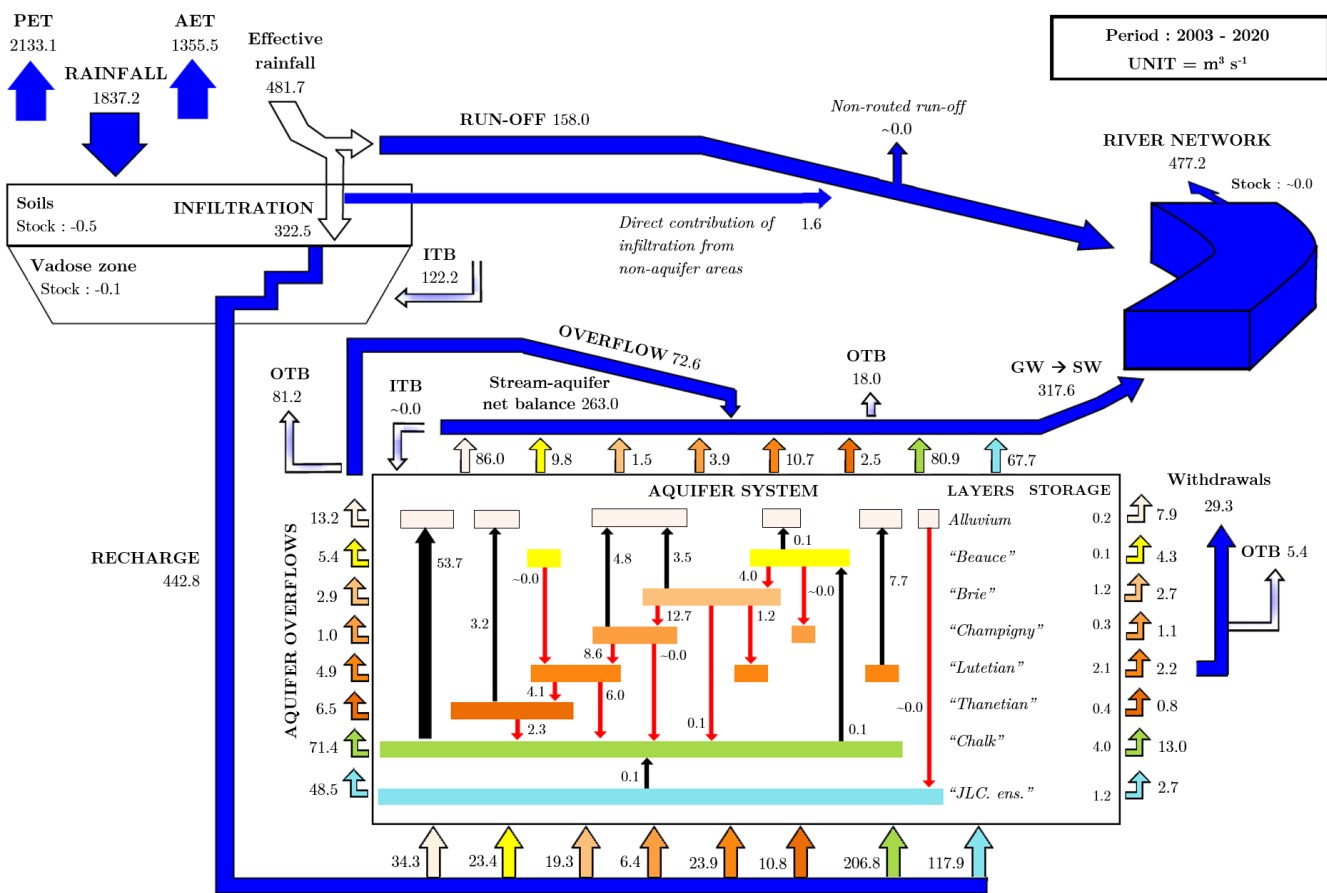

**Figure 13.** Summary diagram of the structure and average functioning of the Seine hydrosystem, as simulated by the CaWaQS–Seine application. All flows expressed in inter-annual average values ($m^3\,s^{-1}$), over the 2003–2020 period. "Beauce": Beauce limestones ensemble, "Brie": Brie limestones and Fontainebleau sands, "Champigny": Champigny limestones, "Lutetian": Lutetian limestones, "Thanetian": Thanetian sands, "JLC. ens.": Jurassic and Lower Cretaceous. $ITB$: Infiltration flux for outcropping aquifer units beyond the Seine basin limits (see Figs. 10 and 11). $OTB$: GW fluxes outflowing beyond the basin limits vertical projection on GW system extension.

Finally, a water budget of the Seine basin is established (Fig. 13) over a 17-year cycle that ensures zero storage variations (Flipo et al., 2012).

A very large fraction (73.8 %) of rainfall is converted to AET (average value of 565 $mm\,a^{-1}$). The remaining effective rainfall fraction, *i.e.* , 26.2 % (average value of 201 $mm\,a^{-1}$) is divided into infiltration toward the aquifer system and runoff, representing 17.6 % (135 $mm\,a^{-1}$) and 8.6 % (66 $mm\,a^{-1}$) of rainfall respectively. An infiltration flow of 443 $m^3\,s^{-1}$ (145

$mm\,a^{-1}$) transits through the unsaturated zone (of approx. 96,200 $km^2$), acting as a recharge of outcropping areas of aquifer layers.

**Table 3.** Values of main components of simulated water budget, in $mm\,a^{-1}$. Mean values over the 2003–2020 period, aggregated at the scale of (*a*) Seine and its main tributary river watershed limits (see Fig. 2) and (*b*) main geological domains (see Fig. 4). Regarding water balance components, values in parentheses are expressed as a fraction of rainfall. As for the characterization of local anthropogenic pressure, indicator values are expressed as the ratio between mean inter-annual withdrawn volume of underground water and the aquifer recharge (2003–2020 period). The actual withdrawn volume is specified in parentheses.

| | Main subsector Area [$km^2$] | Rainfall (mm) | PET (mm) | AET (mm) | Infiltration (mm) | Runoff (mm) | Anthropogenic pressure (%) [$km^3\,a^{-1}$] |
|---|---|---|---|---|---|---|---|
| | Aisne [7,904] | 776 | 867 | 564 (0.726) | 155 (0.200) | 58 (0.074) | 4.0 (0.048) |
| | Yonne [10,695] | 849 | 943 | 602 (0.709) | 109 (0.128) | 138 (0.163) | 7.0 (0.082) |
| | Aube [4,551] | 790 | 907 | 567 (0.718) | 142 (0.179) | 81 (0.103) | 3.3 (0.022) |
| | Essonne [1,932] | 652 | 902 | 525 (0.805) | 118 (0.181) | 9 (0.014) | 21.0 (0.048) |
| (a) | Eure [5,991] | 657 | 845 | 516 (0.787) | 118 (0.179) | 22 (0.034) | 12.3 (0.087) |
| | Marne [12,675] | 802 | 884 | 572 (0.713) | 138 (0.172) | 92 (0.115) | 4.9 (0.085) |
| | Oise [16,804] | 758 | 862 | 557 (0.735) | 153 (0.201) | 49 (0.065) | 5.7 (0.148) |
| | Seine (at Paris) [43,162] | 786 | 918 | 578 (0.736) | 123 (0.156) | 84 (0.107) | 8.4 (0.446) |
| | Seine (at Vernon) [63,843] | 773 | 900 | 571 (0.738) | 131 (0.169) | 72 (0.093) | 8.4 (0.704) |
| | Seine (at outlet) [75,499] | 766 | 890 | 565 (0.738) | 135 (0.176) | 66 (0.086) | 8.9 (0.903) |
| (b) | Jurassic & Lower Cretaceous [30,547] | 895 | 895 | 605 (0.676) | 123 (0.138) | 167 (0.187) | 2.1 (0.078) |
| | Upper Cretaceous & Tertiary [64,562] | 724 | 878 | 549 (0.759) | 151 (0.208) | 24 (0.034) | 10.3 (1.005) |
| | Whole aquifer system [96,204] | - | - | - | 145 (-) | - | 7.8 (1.094) |

Anthropogenic withdrawals from the aquifer system account for 7.8 % of the total aquifer recharge (approx. 1.09 $km^3\,a^{-1}$). As discussed in section 4.3, the exfiltration regime from the aquifer system to the hydraulic network largely dominates river–

aquifer exchanges, as shown by the positive net exchange values in figure 13. Within the basin limits, flows drained from the underground system are, on average, responsible for 67 % (318 $m^3\,s^{-1}$) of the Seine River discharge at the outlet of the basin (477 $m^3\,s^{-1}$). A complementary input to the river network is composed of runoff contributing to 160 $m^3\,s^{-1}$, which accounts for 33 % of the river discharge at the outlet of the basin.

One main advantage of distributed models is the ability to calculate water budgets also for sub-basins of the regional Seine basin. More details about the spatial distribution of the water budget over the Seine basin are provided in Tab. 3.

## 5  Discussion on the relevance of the proposed methodology across scales

The step-wise fitting methodology depends on a distributed minimalist reduction of hydrological parameters, which is performed using the HYMIT methodology (Schuite et al., 2019) coupled with optimization of the distributed parameters of the forward model CaWaQS3.02 using mostly MCMC optimizations or set up by analogy. HYMIT is a very powerful method that may provide a consistent view of minimalist hydrosystem hydrological functioning at various scales given that its fundamental hypotheses are fulfilled; therefore, it is a useful companion for adjusting various hydrological models from the catchment scale to the continental scale (Flipo et al., 2014).

### 5.1  Fundamental hypotheses behind the step-wise methodology

The proposed methodology relies on two main assumptions that have to be fulfilled before it can be applied:

- the stationarity of hydrological signals over the period of time for which the study is run, especially for performing the transfer function analysis with the HYMIT, which involves effective rainfall and river discharges;

- the overlapping of surface and subsurface catchments.

Before discussing these hypotheses and their validity across scales, let us introduce the instantaneous equation for the water budget at the surface basin scale:

$$p(t) - aet(t) = q_{out}(t) - q_{bound}(t) + \Delta s_{riv}(t) + \Delta s_{sub}(t) \tag{3}$$

where $p(t)$ [$m^3\,s^{-1}$] is the precipitation rate, $aet(t)$ [$m^3\,s^{-1}$] is the actual evapotranspiration rate, $q_{out}(t)$ [$m^3\,s^{-1}$] is the river discharge at the basin outlet, $q_{bound}(t)$ [$m^3\,s^{-1}$] is the incoming subsurface flux through the basin boundary, $\Delta s_{riv}(t)$ [$m^3\,s^{-1}$] is the water storage variation in the river network and $\Delta s_{sub}(t)$ [$m^3\,s^{-1}$] is the water storage variation in the subsurface, both considered over time interval $\Delta t$ [$s$].

The first step in the step-wise methodology relies on the estimate of $aet$, which is done by simplifying Eq. (3). The two hypotheses make this step possible on whatever scale it is applied: from small catchment scale to continental scale. The second step of the step-wise methodology also relies on these two assumptions.

### 5.1.1 Overlapping of surface and subsurface catchments

The inflow or outflow through the limits of the surface topographical basin projected onto the subsurface groundwater domain is a quantity that is neither observed nor measured, whatever the scale of the system of interest, making the estimation of the term $q_{bound}(t)$ of Eq. (3) almost impossible. Assuming that the surface and subsurface watersheds overlap (Tóth, 1962), the subsurface fluxes at the basin boundaries (lateral and bottom) can then be neglected (*i.e.*, $q_{bound}(t) = 0$).

This hypothesis holds in sedimentary basin environments where fractured or karstified areas are not preponderant. Tóth (1962) shows that in sedimentary basins, most of the river water fluxes originate from shallow subsurface flows and that piezometric heads are strongly correlated with surface topography. Thus, on a sedimentary hydrosystem, all the watersheds of gauged tributaries of the main hydrographic network can be considered as sub-hydrosystems.

### 5.1.2 Stationarity of hydrological signals

From a theoretical standpoint, signal stationarity is ensured when its mean and variance remain constant over time. The presence of an underlying trend in hydrological records causes a variation in mean, whereas multi-scale natural fluctuations may affect variance stability. Signal stationarity is particularly important for the two first steps of the nested fitting approach presented in this paper.

Large-scale climate oscillations induce large periodic variations of both surface and subsurface water stock and fluxes (Flipo et al., 2012; Massei et al., 2010). In the absence of long-term trends, these quantities are stationary over major pluri-annual hydro-climatic periods, such as the North Atlantic Oscillation (Flipo et al., 2012). Over such a climate period, the time integrals of storage variations in subsurface and surface compartments are therefore negligible ($\int \Delta s_{sub}(t)\, dt \approx 0$ and $\int \Delta s_{riv}(t)\, dt \approx 0$).

Integrated over a major hydro-climatic period, Eq. (3) yields:

$$\overline{P} - \overline{AET} = \overline{Q_{out}} \tag{4}$$

where $\overline{P}$ $[m^3\, s^{-1}]$, $\overline{AET}$ $[m^3\, s^{-1}]$ and $\overline{Q_{out}}$ $[m^3\, s^{-1}]$ are the averaged precipitation rate, actual evapotranspiration, and total discharge at the basin outlet, respectively.

The first step in the methodology hence requires the identification of stationary time windows for any given hydrosystem, given the *a priori* knowledge of its pluriannual modes of climatic variability (in the case of the Seine basin, 17 years). Over such a period for which hydrological signals are stationary, $\overline{AET}$ can therefore be estimated from classic hydrological data that are either measured in situ or spaceborne.

For the second step, namely, the application of HYMIT, first-moment stationarity is routinely forced by detrending signals prior to frequency-domain transformation. By taking into account only the longest and most complete hydroclimatic datasets, which is necessary in order to benefit from the statistical power of HYMIT analysis, we also maximize variance stability over time, owing to the large ratio of short-term fluctuations to long-term oscillations.

## 5.2 Estimating hydrosystem inner fluxes across scales

We developed herein a step-wise fitting procedure that leverages the HYMIT method and that displays unprecedented performance especially for simulating hydrosystem inner fluxes, which are the most uncertain in many current calibrated/validated hydrosystem models, either Land Surface or Hydrological Models (LSMs and HMs) (Samaniego et al., 2017).

One explanation could be the equifinality that stems from large uncertainties in the identification of parameter values (Beven and Binley, 1992; Beven, 2006; Ebel and Loague, 2006). Not only the distribution of actual evapotranspiration or soil moisture are potentially miscalculated (Stisen et al., 2011; Rakovec et al., 2016) but also the distribution of river–aquifer exchanges remains uncertain along a river network at the watershed scale (Barclay et al., 2020). It is therefore of primary importance to further develop fitting methodology for HMs and LSMs (O'Neill et al., 2021).

Although it remains important to check the model ability to reproduce physical processes based on a simplistic case study (Maxwell et al., 2014; Tijerina et al., 2021), it is still very preliminary in terms of model development and not sufficient to meet the challenge of the hyper-resolution that imposes hydrological predictions to be relevant "everywhere" on earth, either at the outlet of large river basins or at the local catchment scale of a few $ha$ or $km^2$ (Wood et al., 2011). Using multi-scale basin outlets over a continental scale as a basis of the objective function improves LSM performance significantly but does not prevent them from providing divergent results for various resolutions (Rakovec et al., 2019). It, nevertheless, narrows the issue of equifinality. Also, introducing supplementary hydrological fluxes in addition to river discharges to objective functions usually improves model performance (Baroni et al., 2019). HYMIT has the advantage of providing multi-scale estimates of hydrosystem parameter values that are the basis of the estimation of inner fluxes such as regional-scale runoff expressed by beta times the effective rainfall. It provides such estimates in a fully physically consistent framework that is based on a Fourier domain minimalist reduction and therefore provides invaluable additional data to shape objective functions that reduce equifinality drastically.

Important findings about equifinality were recently reported. First, Cuntz et al. (2015) argued that equifinality may be an artifact of flawed calibration procedures that focus on non-sensitive parameters and unsuitable objective functions. Just as important are the findings of Samaniego et al. (2017), who show that most of the state-of-the-art LSMs and HMs do not fulfill flux-matching conditions across scales. As a consequence, they do not have consistent hydrologic parameter fields across scales and more problematic is that their parametrization at large continental scale is still unresolved. One way to overcome this issue is then to use multiscale parameter regionalization (MPR), as proposed by Samaniego et al. (2010) and further improved by Kumar et al. (2013), for LSMs as successfully demonstrated by Mizukami et al. (2017) and Samaniego et al. (2017). MPR relies on the identification of intrinsec hydrogeophysical parameters on which scaling operators are based to ensure flux continuity over scales. Combining this powerful calibration technique with the estimates of inner fluxes of hydrosystems provided by HYMIT could be the next step toward improving the robustness of LSMs and HMs and therefore their prospective power for assessments of climate change impact. The nested hydrological fitting step-wise methodology that assumes the dependency of parameter fields on boundary conditions, especially boundary conditions of fluxes, that was used on the Seine basin could be

adapted to identify intrinsic parameter fields such as formulated in mHM rather than HRU-based parameters as is the case in CaWaQS3.02 for the surface compartment.

Finally the value of the step-wise methodology was demonstrated at the regional scale. As discussed above, its potential at the continental scale, which can be viewed as a collection of regional systems (Flipo et al., 2014), seems important and needs to be tested. It would also be challenging to evaluate in a heuristic way until which fine scale the hypothesis of the overlapping of surface and subsurface catchments holds. The results of this evaluation could provide a breakthrough in hydrological modeling of the critical zone, since most of the data are acquired in fine-scale catchments, for instance, at the scale of critical zone observatories (Gaillardet et al., 2018). The Orgeval catchment located in the Seine basin corresponds to a hydrosystem on which **(i)** many long-term high-frequency hydrological datasets exist (Mouhri et al., 2013; Floury et al., 2017) and **(ii)** such an evaluation could be carried out especially to elucidate water pathways that constitute fundamental information for the understanding of the biogeochemical behavior of hydrosystems (Floury et al., 2019; Tunqui Neira et al., 2020).

## 6   Conclusions

A step-wise methodology for fitting HMs and LSMs is proposed and demonstrated on the Seine basin. It leverages the analysis and the determination of a distributed minimalist hydrological parameter set in the frequency domain with the HYMIT methodology (Schuite et al., 2019) that serves as a basis for the estimation of external and internal water fluxes in the time domain with CaWaQS (Flipo et al., 2021a) at both the regional and the territory scales.

The methodology is exemplified with the Seine basin, offering for the first time a very detailed picture of the basin functioning as a whole but also at the scale of territories. All hydrological fluxes are estimated in a consistent way between the frequency and the time domains, from classic ones such as AET, river discharges, to more challenging ones such as GW contribution to river discharge, or exchanges between aquifer units, via the share between fast runoff and slower infiltration. To the authors' knowledge, it is the first time that such a characterization of hydrological system behavior of a regional-scale river basin is proposed, and which reproduces the observations fairly well in both time and frequency domains.

This development paves the way for significant breakthroughs in hydrological modeling of systems over a large range of scales, from small catchments to regional/continental river basins.

*Code and data availability.* CaWaQS3.02 is available under an Eclipse Public Licence 2.0 in the following zenodo deposit (Flipo et al., 2022b). MATLAB-based HYMIT scripts are available under an Eclipse Public Licence 2.0 in the following zenodo deposit (Schuite, 2022). Hydrological datasets, especially the one associated with Figs. 8, 9c&d, 10, 11 and 12, are available in the following zenodo deposit (Flipo et al., 2022a).

## Appendix A: CaWaQS water balance and AET calculation

As previously stated, ensuring a robust water balance parameterization is crucial. Numerous and complex elementary mechanisms govern water flows at the soil surface and in its sub-surface layers. These are often associated with parameters difficult or even impossible to acquire at the regional scale. This acts in favor of the use of a more global conceptualization such as a reservoir model (Girard et al., 1980), as implemented in CaWaQS (see section 2.2, Fig. 3). This approach allows a representation of Hortonian flows (Horton, 1933). Soils are associated with a given infiltration capacity producing surface runoff when exceeded. Here, a set of four reservoirs (cf. Fig. A1) is used to ensure water balance calculation as well as water release dynamics to the surface and underground compartments. Although not being strictly measurable *in situ*, production-function parameters are in relationship with actual physical quantities representative of the soil system state.

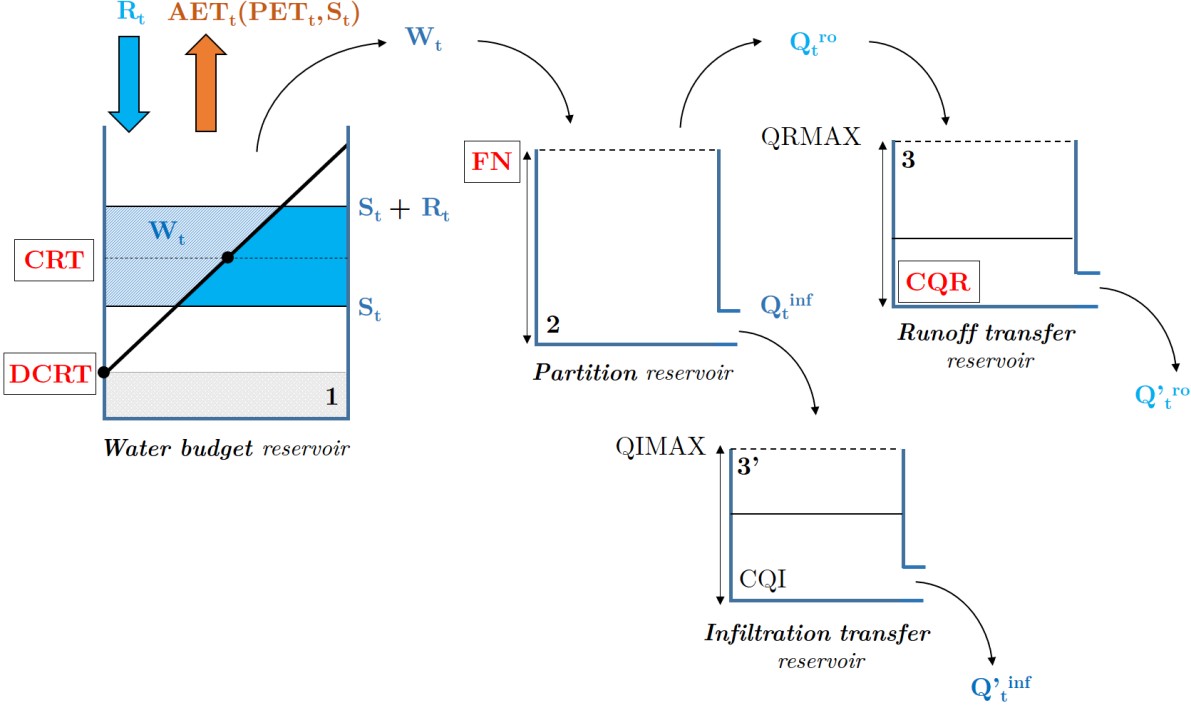

**Figure A1.** Schematic illustration of a CaWaQS production-function (cf. Fig. 1). Reservoir-based conceptualization used for water balance and AET calculation in CaWaQS3.x. Time-step dependant values use the subscript $t$. CaWaQS calibration parameters are listed using block letters. Automatically calibrated parameters using the HYMIT-MCMC method are mentioned in red.

CaWaQS water balance calculations rely on :

- a budget reservoir (cf. Fig. A1-**1**), based on daily values of total rainfall $R_t$, potential evapotranspiration $PET_t$ and soil water storage $S_t$, according to equations (A1) and (A2). The amount of available water $W_t$ set for circulation in the hydrosystem is determined according to the storage $S_t$ value and in relation to the DCRT and CRT levels of the

soil reservoir. DCRT represents the minimum value of the water stock in soil, below which no water is available. It regulates the role of first rainfalls after a drought period. CRT is the average value of the water stock in soil. $AET_t$ increases according to this parameter, which thus conditions global water balance. Both are expressed in $mm.d^{-1}$. Actual evapotranspiration $AET_t$ is calculated based on the remaining reserve after subtraction of the $W_t$ quantity up to the $PET_t$ value.

$$AET_t = min(S_t + R_t - W_t, PET_t) \tag{A1}$$

with :

$$
\begin{aligned}
W_t &= max(S_t + R_t - RMAX, 0) + \frac{DR_t(2\,RBA_t + DR_t)}{4\,(CRT - DCRT)} \\
RMAX &= 2\,(CRT - DCRT) + DCRT \\
DR_t &= max(0, RHA_t - RBA_t) \\
RHA_t &= min(S_t + R_t, RMAX) - DCRT \\
RBA_t &= max(DCRT, S_t) - DCRT
\end{aligned}
\tag{A2}
$$

- a fractioning reservoir (cf. Fig. A1-**2**), which distributes the $W_t$ quantity into runoff ($Q_t^{ro}$) and infiltration ($Q_t^{inf}$) fluxes, by comparison with a threshold value (FN), which represents the maximum infiltration rate over a given time-step,

- two transfer reservoirs (cf. Fig. A1-**3** and A1-**3'**) regulating release dynamics of infiltration and runoff flows. They allow the computation of direct and delayed water flows ($Q_t^{'ro}, Q_t^{'inf}$) using respectively the CQR and CQI recession constants of the runoff and infiltration reservoirs. Respective overflow levels of the runoff and infiltration reservoirs are labelled as QRMAX and QIMAX.

*Author contributions.* NF and JS conceived the step-wise methodology. JS developed the MATLAB scripts of the HYMIT method (Schuite, 2022) and performed the first proof of concept of the step-wise methodology. NF managed the development of the CaWaQS software since 2000. NF and NG developed with other collaborators the CaWaQS3.02 software (Flipo et al., 2022b). NG set up the coupled model and performed the calibration, all the calculations presented in this paper as well as pre-processing of the large datasets and post-processing of model outputs. NG, NF and JS analysed the results. NF, NG and JS co-authored the paper and NG designed the figures. NF secured the pluri-annual funding of the work and the management of the project.

*Competing interests.* The authors declare that no competing interests are present.

*Disclaimer.* Results and data of the present study are freely available. However any further usage of such will be made at the user own responsibility. The authors of the present paper can not be considered responsible for either any flaws appearing later in the data, nor for their interpretation or usage by third parties.

*Acknowledgements.* This work is a contribution to the PIREN-Seine research program (www.piren-seine.fr), which is part of the French e-LTER Zone Atelier Seine. The study was also partly funded by the Seine Normandy Water Agency (AESN) through the AQUIVAR and AQUIVAR+ projects. The authors kindly thank Baptiste Labarthe, Agnès Rivière, Fulvia Baratelli, Mathias Maillot and Deniz Kilic for their contribution to the development of the CaWaQS3.02 software. Daily volume records, for all four reservoirs since their respective construction dates, were kindly provided by the *Etablissement Public Territorial de Bassin Seine Grands-Lacs*, basin institution for river water flow regulation. GW withdrawals were compiled from long-term data provided by the Seine-Normandy Water Agency, over the 1994–2012 period. Complementary data were collected using the BNPE database (https://bnpe.eaufrance.fr/, French Water Withdrawals National Database).

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
