# Peer review of "Regional coupled surface-subsurface hydrological model fitting based on a spatially distributed minimalist reduction of frequency-domain discharge data"

_Geoscientific Model Development, 2022_

## Author Response (AR2)

Dear Editor,

We thank you for your comments on our manuscript that we addressed point by point in blue hereafter

Dear Authors, Thank you for submission of your revised manuscript.

I realized that both reviewers requested for explanations regarding clarity of some aspects in your manuscript and unfortunately you did not elaborately attend to the several comments of both reviewers as required.

We thank you for your remark to which we only partly agree to the extent that we did address the majority of the reviewers' comments, that were not related to either the innovative methodology we proposed in this paper, nor to the soundness or quality of our analysis or very detailed results, but mostly on explanations of a few details of the paper (first reply to reviewers in appendix of the present reply).

Albeit you recommended minor revisions, we reread carefully our paper and finished to exhaustively reply to the four remaining questions that will hopefully help clarify a few technicalities of the paper. Before replying point-by-point to your four main comments, we first reply to your last statement that is more general.

Unfortunately, your responses to the above comments of both reviewers comprised your defenses in line with the length of the paper as well as the claim for readership. I do agree with the reviewers especially regarding the need to ensure your work is clear for the ease of reproducibility. Simply referring to other people's work on methods which you used in your manuscript is not good enough for readers to fully understand and apply the method in your paper.

We are concerned by your mentioning of "Simply referring to other people's work on methods....", which is, to our opinion, a misunderstanding between us. We are reticent to provide more details on Schuite et al. for ethical reasons, and more specifically for not being accused of self plagiarism since J. Schuite (the first author of Schuite et al. (2019)) is a co-author of the current paper, and the first author of the current paper is the main co-author of Schuite et al. (2019). This is why, we synthesized the quintessence of HYMIT, which is essential for the current paper in a long explanation, which is to our opinion sufficient for the understanding of the current paper. However, we understand your point and propose a mitigation measure to balance this ethically sensitive comment.

1. R1 i) "some description of how the AET was computed"
   As suggested we explain the AET computation in appendix.

2. R1 ii) "a need for the use of the Nash model which has to be discussed more in detail with the relevant literature"
   We agree that our first description was too short. We explicitly added an explanation, l.185-193, of the two processes associated to flow in the unsaturated zone: diffusion in the porous medium matrix and preferential flow. Even though more and more acknowledge it as a substantial proportion of aquifer recharge, the preferential flow remain of secondary importance at the regional scale. The usage of a gamma distribution (equivalent to

the nash reservoir cascade) is still today very common and relevant. Associated references, mostly recent, have also been added. Also an explicit mention to observed unsaturated zone effect on discharge data, compatible with our formulation was added l. 258-260

3. R1 iii) "description and interpretation of hydrological time series with respect to Fourier Transforms", R2i)"provide a pictorial description of HYdrological MInimalist Transfer function method", R2ii)"state and explain the link between parameters that control the shape of the transfer function"

With our concern on self plagiarism in mind, we propose the addition of a pictorial description of HYMIT in the paper, with the example of a transfer function of effective rainfall into discharge and by visually explaining how key parameters control its main shape and specific features (answer to R2i).

Along with this new figure 5, we added in the text l. 246-258 the relevant description of each parameter's influence on the transfer function, supplementing our initial intent to present HYMIT's parameter in paragraph 2.3 (answer to R2ii).

Additionally, we rephrased and expanded the description of the method used to calculate and interpret experimental transfer functions with HYMIT, which now reads l.237-240: "*The ratio of the power spectral density (PSD) of the naturalized river discharge over the one of the effective rainfall gives an experimental transfer function (ETF) (Pedretti et al. 2016, Schuite et al. 2019). The PSD of a signal is obtained by squaring the module of its Fourier transform, which is computed using a classical fast Fourier transform algorithm ('fft' function in MATLAB).*" (answer to R1 iii).

Finally, to address the concerns of both reviewers on the meaning of HYMIT's shape, as well as the interpretation of hydrological signals in the frequency-domain, we rephrased and expanded the paragraph l.259-264 which now reads: "*Schuite et al. (2019) demonstrated that HYMIT is sensitive to all parameters on both a synthetic case study and real data, but the shape of the TF is ultimately governed by a complex interplay between physical properties of hydrosystems and the characteristic flow time scales they induce in each hydrologic compartment. For instance, the central energy depression in the TF, known to appear in the presence of a strongly inertial unsaturated zone, was observed for the Essonne watershed but not for the Aube watershed, yet both nested within the Seine basin (Schuite et al. 2019).*"

We are now confident that these modifications and additional elements provide the right amount of detail for the interested readers to understand and reproduce our work, while avoiding the pitfall of self plagiarism and while not focusing too much on technicalities that are not, as you know, at the center of the novel stepwise fitting methodology we propose in the paper.

4. R1iv) "a need for more comparative analysis regarding how you relate

your work with those of others not necessarily from your group"
Here we have to reiterate the expression of our fundamental incomprehension of this comment as far as the references section was already 8 page long, meaning that we are mostly referring to other groups research. However we fully re-read our paper and, on the one hand, identified a few missing references, and on the other hand, identified new references from the year, that we added in various sections of the paper. The bibliography is now 9.5 page-long which hopefully should meet the reviewer expectations.

Therefore, I recommend that you take time, revise your manuscript while carefully addressing the various comments of both reviewers 1 and 2. If your worry is on the length of the paper (since it is not a description paper), I recommend that you submit the relevant text in response to the comments of the reviewers as supplementary material to your manuscript.

I hope so much in your cooperation to enable me reach a final decision on your manuscript.

Thanks

We sincerely hope that the hereby response to your concerns and associated modifications and additions in the paper will clarify the few technicalities raised by the reviewers and that your final decision will favour the publishing of our stepwise fitting methodology, that should inspire the community of hydrological and hydrometeorological modellers.
With our best wishes for the end of the year.
Nicolas Flipo on behalf of the co-authors

**appendix: first point-by-point reply to reviewers' comments**

We first want to thank the two anonymous reviewers for having taken time to review our manuscript and their feedback that requires minor revisions of our document. Hereafter is our point by point reply to those suggestions.

**Reply to RC1's comments**

Line 63

(ii) instead of (iii)

Done

Line 86

mHm is more an hydrological, rathe than LSM, better cite CLM, used by Kollet who is cited before.

OK, we cited CLM with reference to the latest publication of Lawrence et al. 2019, as well as O'Neill et al 2021 that was already referenced in our bibliography. We also added an explicit quotation for hydrological models of various complexity with mHM and GR.

Line 161

Please add "The following" before expression (2) because I was looking for a previous expression.

There was indeed a syntaxe issue. Lines 160-164 were reworded

Figure 2 and lines around 156

I wonder what velocity is for almost 200 km (between Auxerre and Paris) in about 5 hours, 40 km/h? I think that a more detailed discussion is necessary regarding the concept of travel, transfer and concentration time in light of the many works in the last years. Especially if we also deal with matter and quality issues. Between the de Marsily blueprint of 1978 and now, a lot of work has been done.

The thesis of Golaz-Cavazzi and the paper in WRR of the first author is a bit too short basis.

The reviewer may have misread the legend of Fig. 2, where the blue labels are expressed in days not hours. Therefore the travel time between Auxerre and Paris is around 5 days. The velocity is then in the order of magnitude of a few decimeters per second, which is correct for such a lowland river. To the authors opinion and to keep the paper as concise as it can be, the calculation is correct and does not require more explanation. No change done.

Line 178

No words are spent for explaining how AET is computed.

Muchlater from figure 5 I guess AET is estimated as a fitting parameter with MCMC, but it has to be described much before, when main fluxes are described.

The paper is a method paper for assessment of models, not a model description paper. The highlight is on the stepwise fitting procedure, what it brings in terms of insights into a regional system inner fluxes estimation, which leads to a better understanding of hydrological functioning, especially the importance of groundwater. Finally the discussion opens door to a generalisation to other types of models. We believe that describing CaWaQS in more details will lose the readership. References are proposed to refer to.

Line 184

Also regarding vadose zone, inherently nonlinear, the use of the Nash model has to be discussed more in detail with the relevant literature. The two parameters can adjust even a wrong model...

We agree that the use of a reservoir cascade for representing the vadose zone may be surprising at first glance, but its usage was validated and adjusted for river discharge gauging station by Schuite et al. (2019) (see lines 236-238), which is a proof of work of this simple concept that is adapted to the regional scale modelling that is performed here.

Line 225

The interpretation of hydrological time series how is dealt with Fourier transforms? There are citations, but some description inside the paper could be helpful.

An entire subsection (2.3 Minimalist reduction of frequency domain hydrological data with HYMIT, lines 215-254) summaries HYMIT and the method. This full page of crucial explanations seems well balanced to us. Readers who

want to dive further in HYMIT concepts and theoretical fundations are warmly invited to refer to Schuite et al. (2019).

Line 239

Beside rainfall, is there any snowfall?

SAFRAN data provides both daily rainfall and snowfall rates separately. Here, they are both summed up and integrated as rainfall CaWaQS inputs. To be more explicit, line 180 and Fig8a now mention 'total rainfall'. Meaning has been added in caption of figure 8, with a comment that explicitly states that the Seine basin is not submitted to significant snowfall. At most there is a few mm on the Morvan ridge, which melt in few days/hours.

Line 249

Streamflow gauging, better specify it

OK, replaced by discharge gauging station

Line 364

Total instead of total

done, also pointed out by RC2

Line 520

The comparison with papers not belonging to the group is restricted to paragraph 5.2, More comparisons are needed.

We thank RC1 to point this very important discussion subsection on the difficulty of estimating hydrosystem inner fluxes across scales, and the potential of hydrological models or models potential improvment if using our stepwise fitting procedure. We are nevertheless not sure to understand RC1's comment properly. The bibliography is already 8 page long, meaning that our group is referring many papers of other groups.

Concluding my review I should like to see also some more description of the technique of frequency domain reduction, I rad all the paper waiting to have some more info.

The paper is already long. The frequency domain reduction is based on the usage of HYMIT to retrieve key paramaters of watersheds that control water inner fluxes. Our method paper is focused on the stepwise fitting procedure, not on the technical aspects of frequency domain reduction, which is the topic of an already published paper of Schuite et al. (2019), to which the interested reader is invited to refer to all along the paper.

**Reply to RC2's comments**

225. A pictorial description containing the input, model(function or equation) and output relation (like Figure 3.) to describe the

HYdrological MInimalist Transfer function method would be really helpful for readers to get a solid overview of this method.

230. As said earlier, it will be really helpful to get link between paremters that control the shape of the transfer function as decribed on line 230

with the transfer fucntion. You can state the equation and explain the paramaters clearly.

Those two comments are related to HYMIT which is, as fully understood by both reviewers, the core of the stepwise fitting procedure on which the paper is dedicated. As already replied to RC1, the description of HYMIT is one page long, and the full description of the frequency reduction would divert the reader to technicalities that are out of the scope of this paper and also, it would be repeating most information that may be found in Schuite et al. 2019. For more technicalities, the reader should refer to the published paper of Schuite et al. 2019

249. Change gauging stations to streamflow gauging stations. (change all other lines with only gauging stations too).

OK, replaced by discharge gauging station everywhere

364. Correct "totla" to total.

Done, also requested by RC1